# How Do Medical MLLMs Fail? A Study on Visual Grounding in Medical Images

**Guimeng Liu**[1][*]  **Tianze Yu**[1][*]  **Somayeh Ebrahimkhani**[1][*]  **Lin Zhi Zheng Shawn**[2,3]

**Kok Pin Ng**[2,3][†]  **Ngai-Man Cheung**[1][‡]

[1]Singapore University of Technology and Design, Singapore
[2]Department of Neurology, National Neuroscience Institute, Singapore
[3]Duke-NUS Medical School, Singapore

## Abstract

Generalist multimodal large language models (MLLMs) have achieved impressive performance across a wide range of vision-language tasks. However, their performance on medical tasks—particularly in zero-shot settings where generalization is critical—remains suboptimal. A key research gap is the limited understanding of why medical MLLMs underperform in medical image interpretation. **In this work**, we present a pioneering systematic investigation into the visual grounding capabilities of state-of-the-art medical MLLMs. To disentangle *visual grounding* from *semantic grounding*, we design VGMED, a novel evaluation dataset developed with expert clinical guidance, explicitly assessing the visual grounding capability of medical MLLMs. We introduce new quantitative metrics and conduct detailed qualitative analyses. Our study across **eight** state-of-the-art (SOTA) medical MLLMs validates that they often fail to ground their predictions in clinically relevant image regions. We note that this finding is specific to medical image analysis; in contrast, prior work has shown that MLLMs are capable of grounding their predictions in the correct image regions when applied to natural scene images. Motivated by these findings, we propose VGRefine, a simple yet effective inference-time method that refines attention distribution to improve visual grounding in medical settings. Our approach achieves SOTA performance across 6 diverse Med-VQA benchmarks (over 110K VQA samples from 8 imaging modalities) without requiring additional training or external expert models. Overall, our work, for the first time, systematically validates inadequate visual grounding as one of the key contributing factors for medical MLLMs' underperformance. Additional experiments are included in the Supp. *Project Page:* https://guimeng-leo-liu.github.io/Medical-MLLMs-Fail/

## 1 Introduction

Generalist multimodal large language models (MLLMs) have demonstrated strong performance across a broad range of vision-language tasks, including visual question answering (VQA) (Wang et al., 2024; Dai et al., 2023; Liu et al., 2023; Chen et al., 2024b; Liu et al., 2024b), image captioning (Li et al., 2023b; Wu et al., 2024), science and mathematical reasoning (Liu et al., 2024d; Zhuang et al., 2025; Shi et al., 2024). Recent efforts have extended these models to the medical domain, with the goal of developing medical MLLMs that can leverage their generalization capabilities to support diverse clinical decision-making tasks.

---

[*]Equal first author. Authors are permitted to list their name first in their CVs.
[†]Corresponding author.
[‡]Project Lead and Corresponding author.

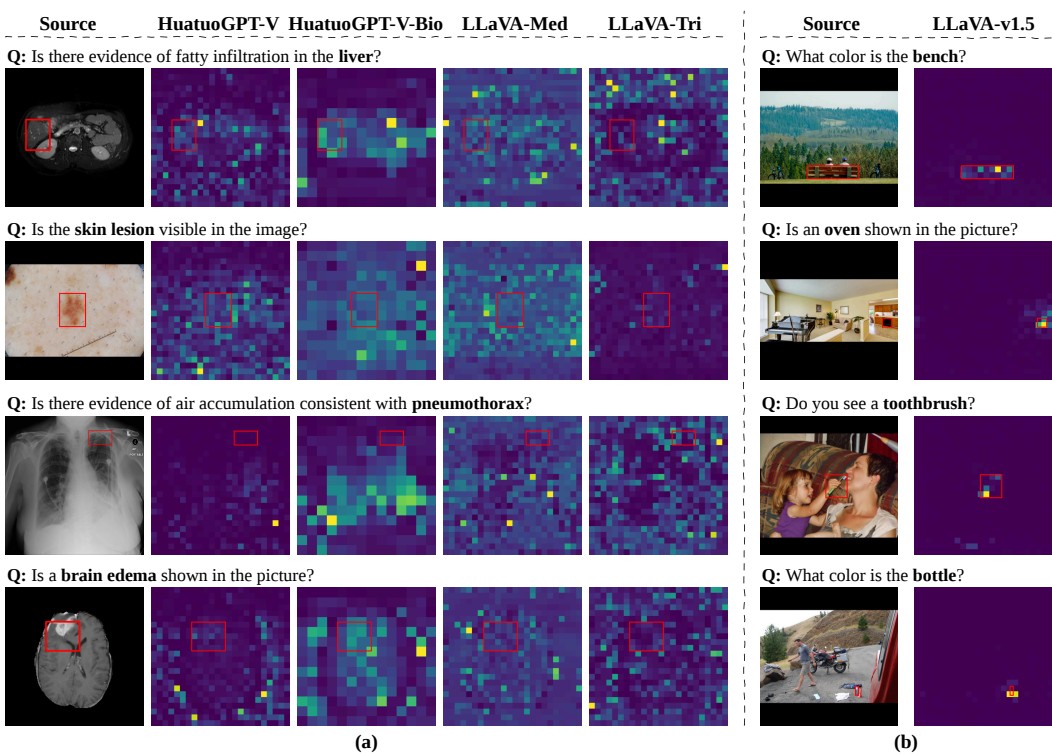

Figure 1: **Visual grounding issues in state-of-the-art medical MLLMs.** (a) Column 1 shows input medical images with expert-annotated ground-truth regions (red boxes). Columns 2–5 display attention distributions from representative medical MLLMs. (b) Column 1 shows natural scene images with annotated ground-truth bounding boxes, and column 2 shows attention distributions from LLaVA-v1.5. For the first time, we systematically validate that state-of-the-art medical MLLMs often suffer from *inadequate visual grounding*—they fail to accurately localize and interpret image regions that are clinically relevant to the question. We note that, in contrast, when applied to natural images, MLLMs are capable of grounding their predictions in the correct image regions (Zhang et al., 2025a). Attention maps are taken from the LLM layers identified as most relevant to visual grounding (see Sec. 2 for details).

**Medical MLLMs.** Recent work has explored extending general-purpose MLLMs to the medical domain, with many approaches focusing on constructing multimodal medical datasets and incorporating external expert models. In Li et al. (2023a), a large-scale biomedical figure-caption dataset is built from PubMed Central to fine-tune LLaVA, resulting in LLaVA-Med. However, its performance in zero-shot settings remains suboptimal and heavily reliant on dataset-specific fine-tuning. HuatuoGPT-Vision (Chen et al., 2024a) leverages GPT-4V to construct a large image-text dataset with refined annotations, but also lacks strong zero-shot generalization. VILA-M3 (Nath et al., 2024) incorporates external medical expert models to assist medical image analysis tasks. Recently, in Xie et al. (2025), the authors introduce MedTrinity-25M, a dataset comprising 25 million medical images, and propose LLaVA-Tri, a model pretrained on this dataset to improve regional focus in medical images. (See Supp. for additional review of related work.)

Despite these advances, most existing medical MLLMs strongly rely on training or fine-tuning with samples from downstream datasets. They continue to underperform on medical VQA tasks in the zero-shot setting—where no downstream task samples are seen during training or fine-tuning—thus falling short of the goal of developing truly generalist medical MLLMs. This raises a key question: *Why do medical MLLMs struggle with medical image interpretation, despite their success in general-domain tasks?*

**Research Gap.** There remains a lack of deeper analysis into the underlying causes of medical MLLMs' suboptimal performance in the important zero-shot setting. Particularly, there is a lack of studies to systematically examine the internal failure modes of these models—particularly in terms of *how* and *where* predictions are derived from visual inputs. Without such analysis, it remains

unclear whether performance limitations stem from a lack of clinical task understanding (semantic grounding) or from an inability to accurately localize and interpret relevant image regions (visual grounding). Advancing our understanding of these failure modes is essential for building robust generalist medical MLLMs for real-world clinical deployment.

Our work underscores the importance of explicitly distinguishing between *semantic grounding* (Lu et al., 2024; Lyre, 2024) and *visual grounding* (Xiao et al., 2024) in medical tasks. This distinction is particularly critical for Med-VQA, which—unlike general-domain VQA—often requires deep domain-specific reasoning. For example, answering a question like "What diseases are included in the image?" requires the model to reason about the anatomical structures and visual features that are relevant to specific pathologies. A model may experience *failure in semantic grounding*—that is, it lacks the medical knowledge to determine *what* to look for. Alternatively, it may experience *failure in visual grounding*—it cannot accurately *localize and interpret* the relevant regions in the medical image, even when it knows what to look for. As medical MLLMs increasingly incorporate large-scale biomedical knowledge to enhance semantic grounding, we argue that visual grounding may emerge as the primary bottleneck limiting further progress.

**Our Contribution.** In this work, we present a pioneering systematic investigation aimed at advancing the understanding of failure modes and the visual grounding capabilities of medical MLLMs (Fig. 1). To disentangle visual grounding from semantic grounding, we co-create a novel evaluation dataset with 3 clinicians, named VGMED, a dataset for Visual Grounding analysis of MEDical MLLMs. VGMED ensures focused evaluation of whether MLLMs can accurately localize and interpret the relevant regions in medical images. We introduce new quantitative metrics and qualitative analyses to assess visual grounding performance—that is, the extent to which model predictions are grounded in clinically relevant visual evidence. *Critically, by using VGMED to evaluate **eight** SOTA medical MLLMs, we reveal for the first time that even the most advanced models frequently rely on spurious or irrelevant regions, highlighting inadequate visual grounding as a pervasive and fundamental failure mode.* We note that this finding is specific to medical image analysis; in contrast, prior work has shown that MLLMs are capable of grounding their predictions in the correct image regions when applied to natural images (Zhang et al., 2025a).

To address this, we propose VGRefine, a simple yet effective inference-time method that improves visual grounding by refining internal attention distributions. VGRefine requires no additional training. Across 6 diverse Med-VQA benchmarks, comprising over 110K VQA samples from 8 imaging modalities (CT, MRI, X-ray, OCT, dermoscopy, microscopy, fundus, ultrasound), VGRefine consistently achieves improved and SOTA performance. Overall, our work offers new insights into the failure modes of medical MLLMs and establishes visual grounding analysis as a necessary diagnostic tool for advancing medical MLLMs in clinical applications.

## 2    INVESTIGATION OF VISUAL GROUNDING IN MEDICAL MLLMS

Despite recent advances, medical MLLMs continue to underperform on complex medical image reasoning tasks, particularly in medical VQA (Hu et al., 2024; Jeong et al., 2024). In this work, we conduct a systematic study to validate that a key limitation lies in inaccurate visual grounding. As a starting point, we analyze attention maps from the model layers most relevant to visual grounding (details on layer selection are provided in Sec. 2.4). As shown in Fig. 1, for medical images, MLLMs' attentions often fail to align with clinically relevant regions.

### 2.1    A NEW DATASET FOR VISUAL GROUNDING ANALYSIS

**Existing medical VQA datasets are ill-suited for visual grounding analysis.** To rigorously evaluate medical MLLMs' visual grounding, we aim to systematically assess the extent to which their outputs are supported by clinically relevant regions of the image (e.g., organs, tissues, or lesions essential for answering a given question). However, existing medical VQA datasets are ill-suited for this purpose, as illustrated in Fig. 2 (a). Many questions, such as "*What diseases are included in the picture?*", can be answered without referencing specific image regions. In contrast, questions like "*What diseases are included in the picture?*" require substantial medical knowledge to determine what to look for, since different diseases, including their stages or subtypes, can manifest with varied and often subtle visual patterns. These patterns are not always well-documented in text and may

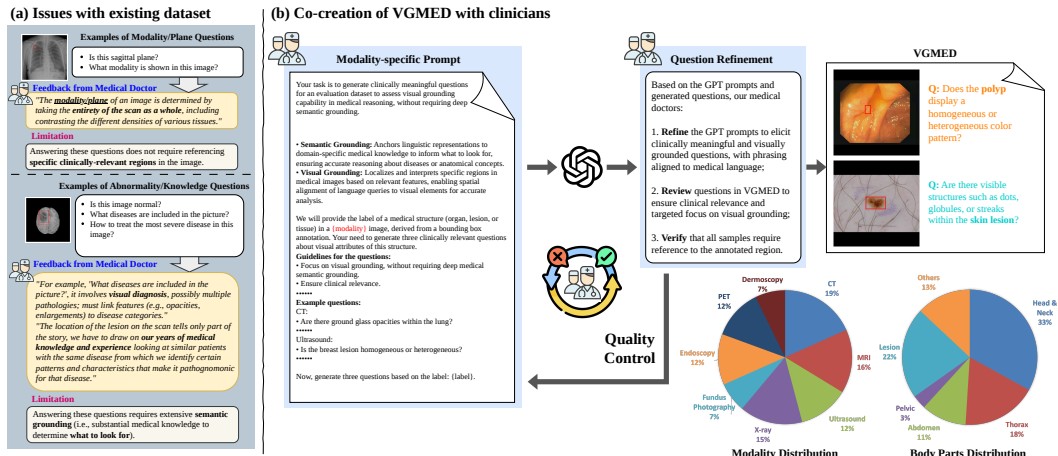

Figure 2: **Co-creation of VGMED with clinicians for visual grounding assessment.** Existing Med-VQA datasets often include questions about image modality or plane, which can be answered without referencing specific image regions. They also contain many abnormality- or knowledge-based questions that require substantial medical expertise to determine what to look for. As a result, existing datasets are not well-suited for analyzing visual grounding. In contrast, our dataset leverages LLM prompting and clinical expert guidance to generate clinically meaningful localization and attribute questions that are explicitly grounded in annotated image regions, enabling rigorous assessment of the visual grounding capabilities of medical MLLMs. **Best viewed in color and with zoom.**

depend on clinical interpretation, making it difficult to determine whether model failures stem from inadequate semantic grounding or from visual grounding alone.

**VGMED: A new dataset for Visual Grounding analysis of MEDical MLLMs co-created with clinicians.** To address this gap, we build an evaluation dataset VGMED, focusing on visual grounding analysis, as illustrated in Fig. 2 (b). VGMED was co-created with three certified medical doctors (general practice, neurology, radiology) to ensure annotation accuracy and clinical relevance, including two senior clinicians with over ten years of experience. One expert also serves as Director (AI and Data Science) at a national medical center. Their contributions included: (1) co-designing GPT prompts to elicit clinically meaningful and visually grounded questions, (2) reviewing and refining all samples for clinical relevance and grounding focus, and (3) verifying that all samples require reference to the annotated region.

Our VGMED dataset is constructed from over 40 publicly available medical image segmentation datasets, with detailed information summarized in Table C.2. The original segmentation masks are converted into bounding boxes to support visual grounding analysis. To ensure diversity across imaging modalities and anatomical regions, we filter 13,962 samples, each consisting of a medical image paired with a ground-truth bounding box. The distributions of modalities and body parts are illustrated in Fig. 2 (b).

For each image–bbox pair, we construct clinically meaningful questions that target specific anatomical or pathological regions, guided by input from clinical experts. This allows us to conduct fine-grained visual grounding analysis. The questions are first generated using GPT-4 and subsequently reviewed and validated by medical professionals. They fall into two categories: *localization* and *attribute* questions. Localization questions inquire about the presence or identification of a specific organ or lesion, whereas attribute questions focus on visual properties such as size, shape, or abnormality (see Fig. 2 for details). GPT-4 is prompted to ensure that questions are both clinically relevant and visually grounded, requiring attention to the entire annotated region. In total, our dataset contains approximately 28K image–bbox–question triplets.

As the reference point, we randomly draw the same number of samples from MS COCO (Lin et al., 2014), using the same question generation pipeline for the evaluation of natural scene images. We include all prompts used in localization and attribute questions generation in Supp I.

**Remark:** Co-created with 3 clinicians, VGMED is a dataset for *evaluation and analysis* of visual grounding in medical domain. The size of VGMED (28K samples) is comparable to datasets typically used in general-domain visual grounding evaluation and studies (see Supp B).

## 2.2 QUANTIFYING MLLMs' VISUAL GROUNDING WITH ATTENTION MAPS

**Measuring MLLMs' visual grounding.** To evaluate how multimodal large language models (MLLMs) ground their predictions in visual evidence, we analyze internal attention maps that indicate which image regions the model attends to. Attention maps are widely used in recent studies to evaluate visual grounding in general-domain MLLMs (Zhang et al., 2025a; Kang et al., 2025; Kaduri et al., 2024). Importantly, Zhang et al. (2025a) demonstrated that attention distributions can reliably capture visual grounding in natural scene images. This enables us to directly compare the visual grounding in medical images and natural scene images.

Alternative grounding indicators, such as gradient-based saliency and causal perturbation, are in principle applicable but are considerably more expensive at scale. Gradient-based saliency methods (Selvaraju et al., 2017; Ismail et al., 2021) require backpropagation for each input, making them substantially more computation-intensive than directly using attention maps from the forward pass. Causal perturbation techniques (Fong & Vedaldi, 2017; Hooker et al., 2019) demand a new forward pass for each perturbed input (e.g., region masking or token removal), which may quickly become prohibitive for large-scale medical grounding analysis.

**Attention maps in MLLMs.** We extract cross-attention weights from the last input text token to each of the $N^2$ image tokens across all $L$ layers and $H$ attention heads of the LLM (Zhang et al., 2024; Kang et al., 2025). For each layer $\ell$ and head $h$, we denote the attention vector as $\alpha^{\ell,h} \in \mathbb{R}^{N^2}$, and compute the average across heads to obtain a per-layer attention map $A^\ell = \frac{1}{H} \sum_{h=1}^{H} \alpha^{\ell,h}$. Then we reshape $A^\ell$ into a spatial attention map of size $N \times N$.

**Attention Ratio (AR).** We aim to measure the alignment from the model's attention map to the ground truth bounding box. For this purpose, we apply attention ratio (AR), defined as the sum of attention inside the ground truth bounding box divided by the average attention inside the bounding box of the same size (Zhang et al., 2025a). Let $A \in \mathbb{R}^{N \times N}$ denote the attention map over image patch tokens, and let $M \in \{0,1\}^{N \times N}$ represent the binary ground-truth mask indicating the annotated region (e.g., bounding box), where $M_{ij} = 1$ if patch $(i, j)$ is inside the region, and 0 otherwise. Formally, AR is defined as $\mathrm{AR} = \frac{\sum_{i=1}^{N} \sum_{j=1}^{N} A_{ij} \cdot M_{ij}}{\frac{\|A\|_1}{N^2} \cdot \|M\|_1}$, where $\|A\|_1 = \sum_{i=1}^{N} \sum_{j=1}^{N} A_{ij}$ and similarly for $\|M\|_1$.

**New metrics to quantify model's attention map alignment.** We note that AR only considers the amount of attention within the bounding box, ignoring how the attention is distributed. Particularly, a uniform distribution of attention within the bounding box region would be preferable, as questions in VGMED are specifically designed to require attention to entire bounding box regions. To take attention distribution into account, we propose to use the Kullback–Leibler (KL) and Jensen–Shannon (JS) divergence, which measure the difference between the attention map and bounding box by viewing them as two probability distributions.

**Kullback–Leibler (KL) divergence.** We compute the KL divergence between the normalized ground-truth mask $\hat{M}$ and the normalized attention map $\hat{A}$ as $D_{\mathrm{KL}}(\hat{M} \parallel \hat{A}) = \sum_{i=1}^{N} \sum_{j=1}^{N} \hat{M}_{ij} \log\left(\frac{\hat{M}_{ij}}{\hat{A}_{ij}}\right)$, where $\hat{A}_{ij} = A_{ij}/\|A\|_1$ and $\hat{M}_{ij} = M_{ij}/\|M\|_1$.

**Jensen–Shannon (JS) divergence.** To obtain a symmetric and bounded divergence metric, we compute the JS divergence as $D_{\mathrm{JS}}(\hat{M} \parallel \hat{A}) = \frac{1}{2} D_{\mathrm{KL}}(\hat{M} \parallel \hat{R}) + \frac{1}{2} D_{\mathrm{KL}}(\hat{A} \parallel \hat{R})$, $\hat{R}_{ij} = \frac{1}{2}\left(\hat{M}_{ij} + \hat{A}_{ij}\right)$. The KL and JS divergences allow us to quantify not only whether the model attends to the correct region, but also how its attention is distributed within that region. A lower divergence indicates better alignment and more consistent attention over clinically relevant areas, offering a complementary perspective to AR.

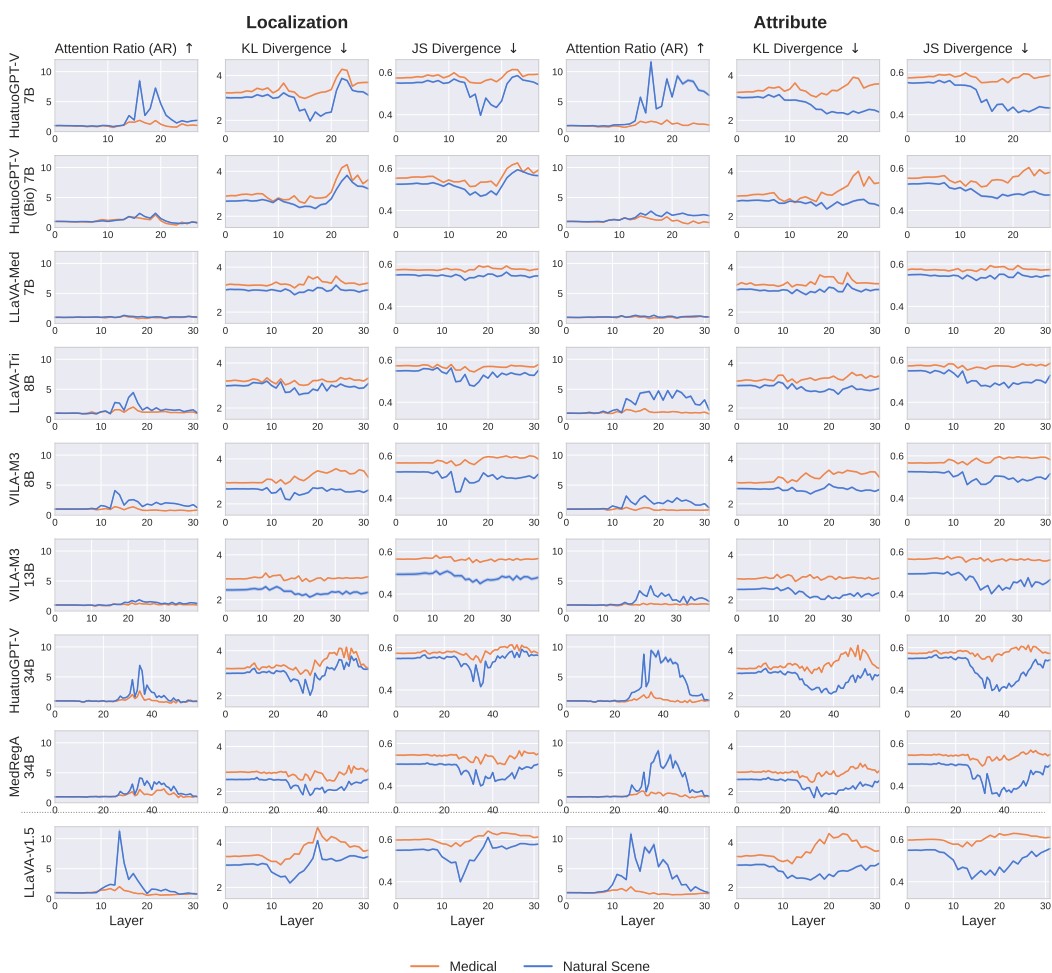

Figure 3: **Medical MLLMs demonstrate suboptimal visual grounding when applied to medical images.** Analysis using our proposed VGMED dataset—designed specifically to assess visual grounding in medical MLLMs—shows that all evaluated medical MLLMs exhibit substantial weaker alignment between their attention distributions and ground-truth annotations on medical images compared to natural scene images (from MS COCO). Additional comparison with general domain MLLM LLaVA-v1.5 on natural images (below the dashed line) further confirms that medical MLLMs consistently exhibit reduced alignment with annotated regions. **Best viewed in color and with zoom.**

## 2.3 EXPERIMENTAL SETUPS

We conduct our analysis on 8 SOTA medical MLLMs, including LLaVA-Med (Li et al., 2023a), LLaVA-Tri (Xie et al., 2025), HuatuoGPT-Vision-7B/34B (Chen et al., 2024a) (abbreviated as HuatuoGPT-V), VILA-M3-8B/13B (Nath et al., 2024), MedRegA (Wang et al., 2025), and a variant of HuatuoGPT-V—referred to as HuatuoGPT-V-Bio—where the original CLIP vision encoder is replaced with BiomedCLIP, a domain-specific encoder trained on biomedical data (see Supp H for details). To analyze attention behavior, we compute the mean attention map across all heads in each LLM layer. Inspired by Zhang et al. (2025a), we normalize the attention map using a reference attention map obtained from the generic prompt: *"Write a general description of the image."*. This normalization helps highlight regions relevant to the specific question.

We also include LLaVA-v1.5-7B (Liu et al., 2024a) results on both medical and *natural scene images*. As a general-domain MLLM, LLaVA demonstrates strong performance and exhibits good visual grounding, with attention distributions that align closely with ground-truth regions (Zhang

et al., 2025a; Kang et al., 2025). This serves as a useful reference point for interpreting attention ratios, KL and JS divergence associated with effective visual grounding.

## 2.4 EMPIRICAL ANALYSIS

**Medical MLLMs exhibit inadequate visual grounding on medical images.** We plot the attention ratio, KL divergence and JS divergence across all LLM layers for all models in Fig. 3. As shown in the figure, all evaluated medical MLLMs demonstrate weaker alignment between their attention distributions and ground-truth annotations when applied to medical images, compared to natural images. This is quantitatively and consistently supported by lower AR and higher values in our proposed KL and JS divergence metrics for measuring attention alignment. These trends persist across most network layers and are consistent for both attribute and localization tasks. Further comparison with LLaVA-v1.5 on natural images reinforces this observation: medical MLLMs show significantly lower alignment with annotated regions, as measured by AR, KL, and JS—highlighting deficiencies in visual grounding for medical image analysis. Note that general-domain model (LLaVA-v1.5) also fails to ground its predictions in clinically relevant regions on medical images and medical VQA tasks, whereas medical-domain models can ground their predictions when applied to natural images and general VQA tasks. This indicates that the grounding failure is not due to model weakness, but is fundamentally specific to the medical domain, consistent with our central findings.

For qualitative analysis, we visualize the attention map from the layer with the lowest KL divergence in Fig.1. Lower KL divergence reflects closer alignment between the model's attention distribution and the annotated regions, indicating that these layers are most relevant for visual grounding analysis. **Comprehensive qualitative analysis and visualization are included in Supp J**

## 3 VISUAL GROUNDING REFINEMENT

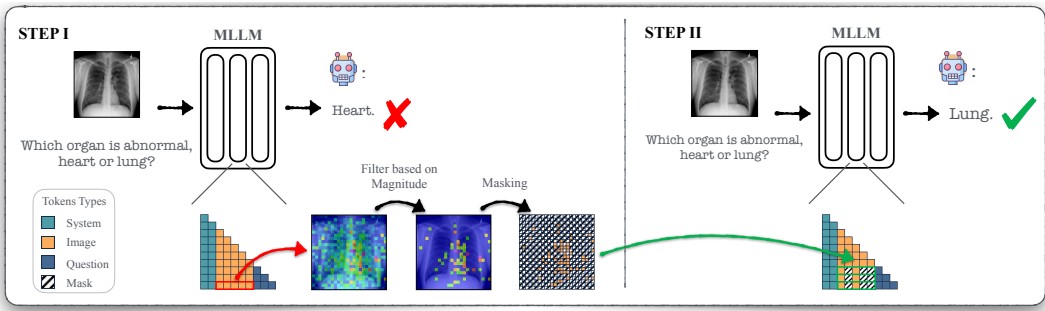

Figure 4: **Illustration of the proposed VGRefine method**: a two-step inference-time method to improve visual grounding in medical MLLMs. In **Step I (Attention Triage)**, we aggregate attention from the model's most visually sensitive heads and suppress low-confident attention, obtaining a binary mask. In **Step II (Attention Knockout)**, we use this mask to refine the model's attention distribution, improving its focus on relevant regions during inference. In the lower triangular attention matrix, each row represents the attention score of a query token to all key tokens.

Our analysis in Sec. 2 suggests that current medical MLLMs attend to clinically-relevant and irrelevant regions. In this section, we propose Visual Grounding Refinement (VGRefine), an inference-time method that enhances visual grounding in medical MLLMs by suppressing attention to clinically irrelevant regions. Specifically, as shown in Fig. 4 our method consists of two steps: 1) Attention Triage and 2) Attention Knockout.

**Step I: Attention Triage — More Focusing on Clinically Relevant Regions.** As illustrated in Fig. 1, medical MLLM's attention maps are often noisy—while they do attend to relevant areas, they also include a substantial focus on irrelevant regions, which diminishes interpretability and precision. To better focus on clinically meaningful regions, we move beyond layer-wise average attention and instead examine visual sensitivity at the head level across all layers. Following the same evaluation in Sec. 2.4, we identify the top $K$ attention heads that most consistently align with visually relevant regions, using our proposed evaluation dataset (Sec. 2.1) and metric (Sec. 2.2).

We then aggregate the attention maps from these selected heads with their average. We suppress low-activation regions based on magnitude of attention, as these are likely to represent irrelevant or noisy attention (see Supp. for further details and motivation). *This results in a sparse attention map with high-confidence.* We convert this filtered attention map into a binary mask $\mathcal{M} \in \{0, 1\}^{N^2}$ by simply setting all non-zero entries to 1 and keeping the zeros unchanged.

**Step II: Attention Knockout — Suppressing Irrelevant Visual Input.** To enhance the visual grounding ability of medical MLLMs, we aim to guide the model's attention toward clinically relevant regions. Intuitively, improving focus on these regions can suppress distractions from irrelevant areas and yield more interpretable predictions. Similarly, recent advances in attention manipulation (Zhang et al., 2024; Geva et al., 2023; Zhang et al., 2025c) have shown that attending to redundant information potentially detriment to prediction as they distract the model's focus, they improve model behavior by preventing attention to uninformative tokens.

Inspired by this, we propose to knock out attention connections between question tokens and clinically irrelevant visual tokens. Specifically, we apply the binary mask $\mathcal{M}$ obtained in Step I to the attention weights $\alpha_q^{\ell,h}$, where $\alpha_q^{\ell,h}$ denotes the cross-attention from the $q$th question token to all visual tokens at layer $\ell$ and head $h$. We compute the masked attention as $\hat{\alpha}_q^{\ell,h} = \alpha_q^{\ell,h} \odot \mathcal{M}$, and use $\hat{\alpha}_q^{\ell,h}$ for the subsequent attention computation in model's forward pass. $\odot$ denotes element-wise multiplication. The masking operation explicitly restricts question tokens from receiving information from irrelevant visual regions at the selected layer. This modification encourages the model to attend selectively to meaningful regions, reducing distraction from irrelevant areas and therefore enhancing models' visual grounding capability.

## 4 EXPERIMENTS

### 4.1 EVALUATION SETTINGS

**Baselines.** We compared two types of open-source models: (1) Medical MLLMs. We evaluated with the latest medical MLLMs, including Med-Flamingo (Moor et al., 2023), RadFM (Wu et al., 2023), LLaVA-Med-7B (Li et al., 2023a), LLaVA-Tri (Xie et al., 2025), MedPLIB (Huang et al., 2025), VILA-M3(Nath et al., 2024), HuatuoGPT-V (Chen et al., 2024a). (2) General MLLMs. We compared with two latest models pretrained on natural scene domain, LLaVA-v1.6-7B (Liu et al., 2024a) and Qwen-VL-Chat (Bai et al., 2023). We include the comparison of larger models in Supp.

**Benchmarks.** We follow the exact evaluation protocol of Chen et al. (2024a). Specifically, we adopt six benchmarks that are designed for biomedical MLLM evaluation, including VQA-RAD (Lau et al., 2018), SLAKE (Liu et al., 2021a), PathVQA (He et al., 2020), PMC-VQA (Zhang et al., 2023b), OmniMedVQA (Hu et al., 2024) (open-access split), and MMMU (Health & Medicine track) Yue et al. (2024). All evaluations were conducted in a zero-shot setting using question templates provided by LLaVA (details in Supp.).

**VGRefine.** We applied our inference-time method on HuatuoGPT-V (Chen et al., 2024a). All experiments are conducted using the same hyperparameters across benchmarks. Specifically, for Step I, we aggregate the attention maps from the top $K$ heads with the highest alignment to visual relevant regions, as measured by KL divergence on our curated evaluation set built using COCO images. This setup prevents data leakage from medical evaluation benchmarks and demonstrates that our method generalizes from natural images to biomedical domains. Low-activation regions are suppressed based on a percentile threshold $p$ over attention magnitude. We discuss the choice of $K$ and $p$ in Sec. 4.4. For Step II we apply the attention knockout only at layer $\ell = 16$ layer, which, according to our analysis in Fig. 3, demonstrates the most relevancy to visual grounding among all the layers.

### 4.2 EXPERIMENTAL RESULTS

We follow exactly the evaluation setup of HuatuoGPT-V (Chen et al., 2024a) to ensure consistency across all benchmarks. Since the original papers of HuatuoGPT-V-7B (Chen et al., 2024a) and VILA-M3-8B (Nath et al., 2024) do not report results on certain benchmarks, we re-evaluate both models under the same zero-shot setting. For models with complete benchmark results available

in their original publications—such as MedPLIB (Huang et al., 2025) and LLaVA-Tri (Xie et al., 2025)—we directly report the official numbers. For all other baselines, we use the results provided in the HuatuoGPT-V paper, as it adopts the same evaluation protocol.

It is important to note that some models include benchmark training sets during pretraining, making zero-shot evaluation unfair. Specifically, VILA-M3 (Nath et al., 2024) and MedPLIB (Huang et al., 2025) incorporate training data from VQA-RAD, SLAKE, PathVQA, and PMC-VQA, and thus are excluded from our zero-shot comparison on those datasets.

**Medical VQA Benchmarks.** Table 1 shows results on four standard medical VQA datasets. Here, we report the closed-ended question accuracy and a weighted average (Avg.) that scales by the number of samples in each benchmark (Additional results are in Supp.). Our inference-time method VGRefine applied to HuatuoGPT-V consistently improves its performance. We observe notable gains of +5.6% on VQA-RAD and +11.3% on PathVQA, with the overall average increasing from 65.3% to 68.4%, outperforming all baselines. These results underscore that enhanced visual grounding contributes to better performance on medical VQA tasks. On the MMMU benchmark (Table 2), VGRefine achieves the highest accuracy across all five sub-domains, increasing the overall average from 45.8% to 47.2%. This demonstrates that enhancing visual grounding at inference time also improves complex multimodal medical reasoning. As shown in Table 3, VGRefine improves performance across all eight imaging modalities, with significant boosts on CT (+7.5%), MRI (+6.4%), and X-Ray (+8.1%) on the OmniMedVQA benchmark. These results confirm the generalizability of our visual grounding refinement across diverse medical imaging tasks. Overall, our method raises average accuracy from 71.3% to 74.4%, demonstrating its robustness and generalizability across a wide range of modalities.

Table 1: Accuracy on medical VQA datasets. To align with the evaluation protocol with HuatuoGPT-V (Chen et al., 2024a), we specifically used the closed-ended subset for evaluation. Evaluation on other subsets in Supp.

| Model | VQA-RAD | SLAKE | PathVQA | PMC-VQA | Avg. |
|---|---|---|---|---|---|
| Qwen-VL-Chat | 47.0 | 56.0 | 55.1 | 36.6 | 48.9 |
| LLaVA-v1.6-7B | 52.6 | 57.9 | 47.9 | 35.5 | 48.5 |
| Med-Flamingo | 45.4 | 43.5 | 54.7 | 23.3 | 41.7 |
| RadFM | 50.6 | 34.6 | 38.7 | 25.9 | 37.5 |
| LLaVA-Med-7B | 51.4 | 48.6 | 56.8 | 24.7 | 45.4 |
| LLaVA-Tri | 59.8 | 43.4 | 59.0 | - | - |
| HuatuoGPT-V-7B | 67.4 | 76.5 | 60.7 | 53.9 | 65.3 |
| VGRefine (Ours) | **71.2** | **76.9** | **67.6** | **56.2** | **68.4** |

Table 2: Accuracy on MMMU Health & Medicine benchmark. **BMS**, **CM**, **DLM**, **P**, **PH** denote Basic Medical Science, Clinical Medicine, Diagnostics & Laboratory Medicine, Pharmacy, Public Health respectively.

| Model | BMS | CM | DLM | P | PH | Avg. |
|---|---|---|---|---|---|---|
| Qwen-VL-Chat | 36.5 | 31.7 | 32.7 | 28.4 | 34.6 | 32.7 |
| LLaVA-v1.6-7B | 40.5 | 36.9 | 32.1 | 32.3 | 26.9 | 33.1 |
| Med-Flamingo | 29.6 | 28.1 | 24.8 | 25.3 | 31.2 | 28.3 |
| RadFM | 27.5 | 26.8 | 25.8 | 24.7 | 29.1 | 27.0 |
| LLaVA-Med-7B | 39.9 | 39.1 | 34.6 | 37.4 | 34.0 | 36.9 |
| LLaVA-Tri | 37.1 | - | 27.8 | - | - | - |
| VILA-M3-8B | 39.3 | 39.7 | 34.0 | 32.1 | 28.7 | 34.0 |
| HuatuoGPT-V-7B | 58.9 | 57.2 | 43.8 | 37.2 | 38.3 | 45.8 |
| VGRefine (Ours) | **59.5** | **59.1** | **45.7** | **38.6** | **39.3** | **47.2** |

## 4.3 HUMAN EVALUATION: VGREFINE IMPROVES TRUSTWORTHINESS

We conducted a blinded study with five experienced clinicians using 20 medical VQA cases from VGMED. Each case presented two attention maps: one from the baseline model and one from the

Table 3: The accuracy of OmniMedVQA within different modalities. Specifically, **FP** denotes *Fundus Photography*, **MRI** denotes *Magnetic Resonance Imaging*, **OCT** denotes *Optical Coherence Tomography*, **Der** denotes *Dermoscopy*, **Mic** denotes *Microscopy Images*, **US** denotes *Ultrasound*.

| Model | CT | FP | MRI | OCT | Der | Mic | X-Ray | US | Avg. |
|---|---|---|---|---|---|---|---|---|---|
| Qwen-VL-Chat (Bai et al., 2023) | 51.5 | 45.4 | 43.9 | 54.0 | 55.4 | 49.5 | 63.1 | 33.5 | 49.5 |
| LLaVA-v1.6-7B (Liu et al., 2024a) | 40.1 | 39.5 | 54.8 | 58.4 | 54.0 | 48.8 | 53.3 | 47.9 | 49.6 |
| Med-Flamingo (Moor et al., 2023) | 34.6 | 33.3 | 27.5 | 26.0 | 28.3 | 28.1 | 30.1 | 33.2 | 30.2 |
| RadFM (Wu et al., 2023) | 33.3 | 35.0 | 22.0 | 31.3 | 36.3 | 28.0 | 31.5 | 26.1 | 30.5 |
| LLaVA-Med-7B (Li et al., 2023a) | 25.3 | 48.4 | 35.9 | 42.1 | 45.2 | 44.0 | 31.7 | 34.4 | 35.8 |
| VILA-M3-8B (Nath et al., 2024) | 60.2 | 35.7 | 51.5 | 56.9 | 51.5 | 51.7 | 65.4 | 46.1 | 53.0 |
| MedPLIB (Huang et al., 2025) | 62.7 | 65.0 | 67.0 | 75.1 | 51.5 | 64.4 | 60.3 | 38.8 | 60.6 |
| HuatuoGPT-V-7B (Chen et al., 2024a) | 62.6 | 80.3 | 67.7 | 86.2 | **71.7** | 74.2 | 74.2 | **79.7** | 71.3 |
| VGRefine (Ours) | **67.3** | **82.4** | **72.0** | **86.9** | **71.7** | **74.9** | **80.2** | 79.5 | **74.4** |

Table 4: Ablation study on the choice of top $K$ heads and $p$ precentile of magnitude-based filtering.

| $K$ | VQA-RAD | SLAKE | PathVQA | PMC-VQA | Avg. | $p$ (%) | VQA-RAD | SLAKE | PathVQA | PMC-VQA | Avg. |
|---|---|---|---|---|---|---|---|---|---|---|---|
| 1 | 68.62 | 75.81 | 64.85 | 53.65 | 68.28 | 30 | 70.78 | 76.84 | 67.55 | 55.70 | 68.22 |
| 2 | 69.78 | 76.63 | 63.58 | 53.90 | 68.21 | 40 | 70.70 | 76.66 | 67.28 | 56.00 | 68.12 |
| 5 | 70.09 | 76.56 | 66.52 | 54.30 | 68.08 | 50 | **71.24** | **76.88** | 67.61 | **56.20** | **68.42** |
| 8 | 70.70 | 76.52 | 66.64 | 55.60 | 68.12 | 60 | 70.70 | 76.66 | 67.28 | 55.65 | 68.05 |
| 10 | 70.86 | 76.81 | **67.67** | 56.05 | 68.34 | 70 | 70.47 | 76.66 | 67.61 | 55.80 | 68.17 |
| 15 | 70.78 | 76.84 | 67.43 | 55.40 | 68.11 | 80 | 70.78 | 76.73 | 67.55 | 55.80 | 68.21 |
| 20 | **71.24** | **76.88** | 67.61 | **56.20** | **68.42** | 90 | 70.55 | 76.34 | **68.11** | 55.50 | 68.20 |

same model after applying VGRefine. The source of each attention map was not disclosed, and their order was randomized. Clinicians were asked which map appeared more clinically reasonable and trustworthy. VGRefine was preferred in 76% of cases, with feedback noting improved focus and reduced noise. These results suggest that VGRefine enhances clinician trust by producing more interpretable visual. See human evaluation details in Supp J.1.

### 4.4 ABLATION STUDIES

Table 4 presents ablations on the number of top attention heads $K$ and the percentile threshold $p$ used for magnitude-based filtering. Performance improves consistently as $K$ increases, with the best average accuracy (68.42%) achieved at $K = 20$, indicating that aggregating more heads helps capture richer grounding signals. For the percentile $p$, the model remains stable across values, with optimal performance also at $p = 50\%$, confirming the effectiveness of moderate filtering in removing noisy regions without discarding relevant information.

## 5 CONCLUSION

In this work, we presented the first systematic analysis of visual grounding in medical MLLMs. Using our clinically guided VGMED dataset and newly introduced metrics, we showed across 8 SOTA medical MLLMs frequent failures in grounding predictions in clinically relevant regions. This failure mode persisted even in recent medical MLLMs and contributed to their underperformance in zero-shot medical image understanding. To address this, we proposed VGRefine, an inference-time attention refinement method to improve medical MLLMs' visual grounding. Across 6 diverse Med-VQA benchmarks, comprising over 110K VQA samples from 8 imaging modalities, VGRefine consistently achieves SOTA performance. We remark that improvements using VGRefine are achieved without retraining or introducing any new medical knowledge. If visual grounding were not a limiting factor, such consistent gains would not occur. Therefore, VGRefine results further support that visual grounding deficiency is a general, widespread issue. Overall, our proposed VGMED helps uncover and confirm inadequate visual grounding, while VGRefine experiments demonstrate its broad prevalence and generalization across different modalities and clinical scenarios. Our findings underscored the importance of grounding-aware analysis to achieve more reliable and generalizable medical MLLMs. **Additional experiments, limitation and ethical consideration are included in Supp.**

ACKNOWLEDGMENT

This research is supported by the National Research Foundation, Singapore under its AI Singapore Programmes (AISG Award No.: AISG2-TC-2022-007); The Agency for Science, Technology and Research (A*STAR) under its MTC Programmatic Funds (Grant No. M23L7b0021). This research is supported by the National Research Foundation, Singapore and Infocomm Media Development Authority under its Trust Tech Funding Initiative. Any opinions, findings and conclusions or recommendations expressed in this material are those of the author(s) and do not reflect the views of National Research Foundation, Singapore and Infocomm Media Development Authority. The work is sponsored by the SUTD Decentralised Gap Funding Grant.

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

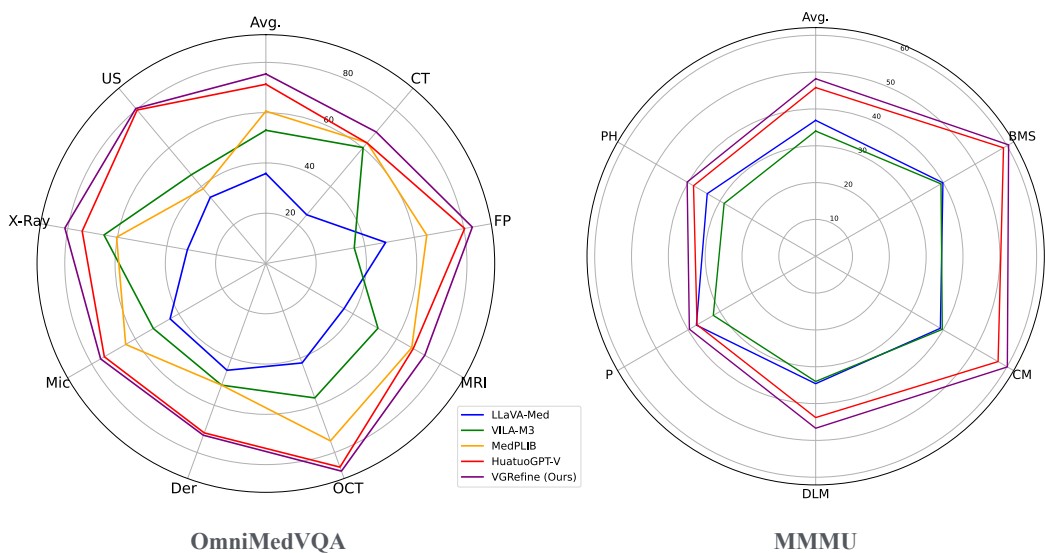

**OmniMedVQA**  **MMMU**

Figure A.1: Our proposed inference-time method VGRefine achieve state-of-the-art performance on OmniMedVQA (Hu et al., 2024) and MMMU (Health & Medicine track) (Yue et al., 2024). Many existing medical MLLMs remain to underperform on medical VQA tasks in the zero-shot setting as shown in this figure, but there is a lack of systematic study to understand the reasons. Compared to existing medical MLLMs, our proposed VGRefine demonstrates consistently stronger zero-shot performance across all modalities and sub-domains, highlighting its effectiveness in mitigating the issue of inadequate visual grounding as revealed in our study.

## APPENDIX OVERVIEW

In this supplementary material, we provide additional experiments, ablation studies, and reproducibility details to support our findings. These sections are not included in the main paper due to space constraints.

Please find the following link for code and other resources: `https://guimeng-leo-liu.github.io/Medical-MLLMs-Fail/`.

## CONTENTS

## A    MORE DISCUSSION ON RELATED WORK

**Medical Multimodal Large Language Models (MLLMs).**    Recent advances in medical multimodal large language models (MLLMs) have focused on leveraging image-text pairs from sources like PubMed central (Zhang et al., 2023b; Moor et al., 2023; Chen et al., 2024a; Li et al., 2023a) and medical textbooks (Moor et al., 2023) to enable generative VQA and medical reasoning. Models such as LLaVA-Med (Li et al., 2023a), MedVInT (Zhang et al., 2023b), Med-Flamingo (Moor et al., 2023), HuatuoGPT-Vision (Chen et al., 2024a), and BioMed-VITAL (Cui et al., 2024) introduce GPT-4 (et al., 2024) generated instruction-following datasets and expert-validated responses to improve medical VQA performance. More recent studies have begun to explore different ideas to improve region awareness in biomedical MLLMs: explicit fine-tuning with additional supervision, such as annotated bounding boxes (Wang et al., 2025; Xie et al., 2025) or segmentation masks (Jeong et al., 2024), along with architectural modifications to support spatial reasoning. For instance, models like MedRegA (Wang et al., 2025) and LLaVA-Tri (Xie et al., 2025) rely on additional datasets. Other recent models focus on scale or domain expertise. VILA-M3 (Nath et al., 2024), for instance, incorporates domain-specific expert models during training, arguing that generic Vision–Language Models (VLMs) lack the fine-grained expertise needed for healthcare. Given their dependence on task-specific fine-tuning (Nath et al., 2024; Xie et al., 2025) and sub-optimal generalization in zero-shot settings (Li et al., 2023a), *it remains unclear whether current medical MLLMs ground their predictions in meaningful visual evidence within medical images.* To our knowledge, no prior work has conducted a comprehensive analysis of visual grounding of medical MLLMs.

**Visual Grounding Analysis in General Domain MLLMs.**    Some recent studies have investigated the internal attention mechanisms of general-domain MLLMs, revealing their potential for implicit visual grounding. Zhang et al. (2025a) demonstrated that MLLMs can identify the correct spatial regions relevant to a given query, even without explicit grounding supervision. They introduce a training-free intervention method (e.g., cropping guided by attention or gradient maps) that enhances performance on general-domain VQA tasks. Broader research into MLLM interpretability has studied how visual information is fused into language representations. Techniques such as causal intervention and cross-modal attention visualization have been employed to offer insights into how vision and language tokens interact through attention mechanisms (Golovanevsky et al., 2024; Zhang et al., 2024; Yu & Ananiadou, 2025; Palit et al., 2023). These studies suggest that middle layers are especially crucial for integrating object-level visual cues with textual context, and that cross-modal attention patterns can encode meaningful spatial alignment signals. However, all of these insights have been drawn from general-domain visual data, such as natural scene images and standard VQA benchmarks. *In contrast, to our knowledge, no prior work has performed visual grounding analysis of medical MLLMs.*

## B    VGMED SCALE COMPARISON WITH RELATED ATTENTION ANALYSIS WORKS

VGMED comprises approximately 28K image-bbox-question triplets, including 14K samples for localization questions and another 14K for attribute questions. The scale of VGMED is larger than or comparable to the number of samples used in the closely related RGB-domain visual grounding (Zhang et al., 2025a; Kang et al., 2025; Kaduri et al., 2024) / attention analysis work (Yang et al., 2025; Jiang et al., 2025; Chen et al., 2025a) (see Table B.1). Unlike RGB datasets that can be constructed by non-experts, our medical datasets require clinical expertise.

Table B.1: Number of samples used in related works.

| Related works | No. of Samples | Data Source |
|---|---|---|
| MLLMs Know (Zhang et al., 2025a) | 4,370 | Text-VQA |
| Your LVLM (Kang et al., 2025) | 1,000 | RefCOCO |
| What's in the Image (Kaduri et al., 2024) | 81 | COCO |
| Hallucination Attribution (Yang et al., 2025) | 1,500 | COCO |
| Devils in LVLM (Jiang et al., 2025) | 2,000 | COCO |
| FastV (Chen et al., 2025a) | 1,000 | 4 VL Tasks |

## C    DETAILED INFORMATION OF DATASETS USED IN VGMED

Table C.2: Detailed information about the 44 datasets incorporated into VGMED. In the "Dataset" column, names such as "StructSeg2019 (Task 1)" represent specific task-based subsets. In the "Anatomical Structures" column, "Others" signifies datasets lacking detailed anatomical data from their original sources.

| Dataset | Modality | Anatomical Structures |
| --- | --- | --- |
| AMOS2022 (Ji et al., 2022) | CT, MR | Abdomen, Thorax, Pelvic |
| ATM2022 (Zhang et al., 2023a) | CT | Thorax |
| AbdomenomenCT-1K (Ma et al., 2022b) | CT | Abdomen |
| BTCV (Igelsias et al., 2015) | CT | Thorax, Abdomen, Pelvic |
| BraTS2013 (Menze et al., 2015) | MR | Head & neck |
| BraTS2015 (Menze et al., 2015) | MR | Head & neck |
| BraTS2018 (Menze et al., 2015) | MR | Head & neck |
| BraTS2019 (Menze et al., 2015) | MR | Head & neck |
| BraTS2020 (Menze et al., 2015) | MR | Head & neck |
| BraTS2021 (Bakas et al., 2017; Baid et al., 2021) | MR | Head & neck |
| CHAOS (Task 4) (Kavur et al., 2021) | MR | Abdomen |
| CTPelvic1k (Liu et al., 2021b) | CT | Pelvic |
| CVC-ClinicDB (Bernal et al., 2015) | Endoscopy | Others |
| Chest_Image_Pneum (Tianchi, 2020) | X-ray | Thorax |
| FLARE21 (Ma et al., 2022a) | CT | Abdomen |
| FLARE22 (Ma et al., 2024) | CT | Abdomen, Thorax |
| HVSMR2016 (Pace et al., 2015) | MR | Thorax |
| ADAM (Task 2) (Fang et al., 2022) | Fundus | Head & neck |
| PALM19 (Fu et al., 2019) | Fundus | Head & neck |
| ISLES (Maier et al., 2017) | MR | Head & neck |
| KiTS2019 (Heller et al., 2020) | CT | Abdomen |
| KiTS2021 (Zhao et al., 2021) | CT | Abdomen |
| LUNA16 (Setio et al., 2017) | CT | Thorax |
| MSD-BrainTumor (Antonelli et al., 2022) | MR | Head & neck |
| MSD-Liver (Antonelli et al., 2022) | CT | Abdomen |
| MSD-Pancreas (Antonelli et al., 2022) | CT | Abdomen |
| MSD-Spleen (Antonelli et al., 2022) | CT | Abdomen |
| CT-ORG (Antonelli et al., 2022) | CT | Head & neck, Thorax, Abdomen |
| PROMISE09 (Bharatha et al., 2001) | MR | Pelvic |
| PROMISE12 (Litjens et al., 2014) | MR | Pelvic |
| SIIM-ACR Pneumothorax (Zawacki et al., 2019) | X-ray | Thorax |
| StructSeg2019 (Task 1) (Huang et al., 2019) | CT | Head & neck |
| StructSeg2019 (Task 2) (Huang et al., 2019) | CT | Thorax, Abdomen |
| TotalSegmentator (Wasserthal et al., 2023) | CT | Head & neck, Thorax, Abdomen, Pelvic |
| Ultrasound Nerve Segmentation (Montoya et al., 2016) | Ultrasound | Others |
| WORD (Luo et al., 2022) | CT | Thorax, Abdomen |
| autoPET (Gatidis et al., 2022) | PET | Pelvic |
| BUSI (Al-Dhabyani et al., 2020) | Ultrasound | Thorax |
| Kvasir-SEG (Jha et al., 2019) | Endoscopy | Others |
| ISIC18 (Task 1) (Codella et al., 2019) | Dermoscopy | Skin |
| ISIC17 (Task 1) (Codella et al., 2018b) | Dermoscopy | Skin |
| ISIC16 (Task 1) (Codella et al., 2018a) | Dermoscopy | Skin |
| SLAKE (Liu et al., 2021a) | CT, MR, X-ray | Head & neck, Abdomen, Thorax |
| PolypDB (Jha et al., 2025) | Endoscopy | Others |

# D    MORE DETAILS OF VGREFINE

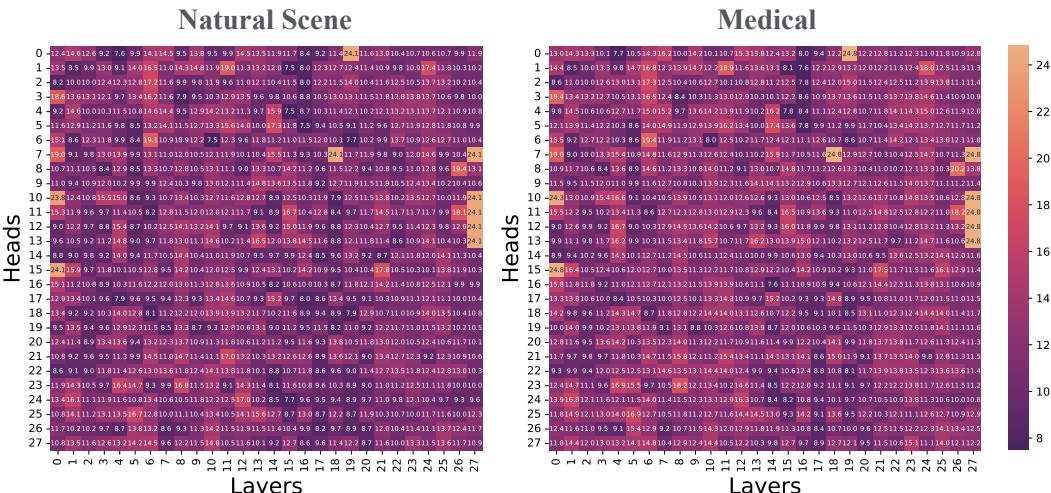

Figure D.1: We conduct an experiment to analyze the alignment between attention distributions from different attention heads and layers and the ground truth annotations in the images. This follows the evaluation setup described in Section 2.3 of the main paper. The medical MLLM evaluated is HuatuoGPT-V-7B. Each cell in the above figures reflects the degree of alignment, measured using our proposed KL Divergence metric (lower is better). *This analysis helps identify the specific heads and layers that are most relevant to visual grounding.* COCO is used for natural scene image analysis, and our dataset VGMED is used for medical image analysis. Interestingly, we find that the attention heads most relevant to visual grounding in natural scene images are often also the most relevant for medical images. However, despite this overlap, the overall visual grounding performance on medical images remains lower than on natural scenes, consistent with the findings presented in Figure 3 (main paper). Based on this analysis, we identify the top $K$ attention heads with the strongest alignment (i.e., lowest KL divergence) and aggregate their attention distributions to compute a refined attention map. *Notably, we select the top $K$ heads using randomly sampled natural scene images from COCO dataset, to avoid data leakage from medical evaluation benchmarks.* This setup also demonstrates that our method generalizes effectively from natural images to the biomedical domain.

In this section, we provide more details of our proposed inference-time method VGRefine (introduced in Sec. 3 of the main paper). Particularly, we discuss how we identify top $K$ attention heads most relevant to visual grounding and leverage their attention distributions in Step I of VGRefine. Fig. D.1 depict the analysis.

We explore attention distributions from different attention heads across all layers, as prior work suggests that individual attention heads in transformers specialize in capturing distinct types of information Voita et al. (2019); Olsson et al. (2022); Gandelsman et al. (2024); Yu et al. (2023); Yang et al. (2025). This motivates us to examine attention at finer granularity to obtain the attention that focusing more on clinically relevant regions.

See details in Fig. D.1. Following the same evaluation setup of Sec. 2.3 of main paper, we assess relevancy to visual grounding of each attention head in HuatuoGPT-V by measuring the alignment between their attention distributions and ground-truth annotations. We perform this analysis using both natural scene images (from MS COCO) and medical images (from our VGMED). The alignment is measured by our proposed KL Divergence ($\downarrow$) as metric.

As shown in Fig. D.1, the visual relevancy patterns are consistent across domains: heads that are relevant to visual grounding in natural scenes also show relative relevancy in medical images, despite exhibiting inadequate visual grounding on medical images compared to natural images (as discussed in Sec. 2.4). Based on this analysis, we select the top $K$ heads with the highest visual grounding relevancy (lowest KL) on natural images and average their attention maps to obtain a

refined attention map. This map is used in Step II to guide the model's improved focus on clinically meaningful areas.

## E    EXPERIMENTS ON OPEN-ENDED MEDICAL VQA

We present additional experimental results on the open-ended questions from the Medical VQA benchmarks. Specifically, we evaluate on VQA-RAD (Lau et al., 2018), SLAKE (Liu et al., 2021a), and PathVQA (He et al., 2020), which include open-ended formats. As shown in Table E.3, our inference-time method consistently achieve better performance across all datasets, demonstrating its effectiveness in enhancing open-ended medical VQA.

Table E.3: Performance comparison on full medical VQA datasets for open-ended medical VQA. We evaluate all models under the zero-shot setting. These results underscore that enhanced visual grounding with our inference-time method VGRefine contributes to better performance on medical VQA tasks. It is important to note that VILA-M3 (Nath et al., 2024), MedPLIB (Huang et al., 2025) and LLaVA-Tri (Xie et al., 2025) incorporate training data from VQA-RAD, SLAKE, and PathVQA, and thus making zero-shot evaluation unfair, and are excluded from our zero-shot comparison.

| Model | VQA-RAD | | | | SLAKE | | | | PathVQA | | | |
|---|---|---|---|---|---|---|---|---|---|---|---|---|
| Metric | BLEU-1 | BERT | OpenRecall | Avg. | BLEU-1 | BERT | OpenRecall | Avg. | BLEU-1 | BERT | OpenRecall | Avg. |
| Qwen-VL-Chat | 28.6 | 63.4 | 27.0 | 39.7 | 28.9 | 52.0 | 33.6 | 38.2 | 18.7 | 45.1 | 9.9 | 24.6 |
| LLaVA-v1.6-7B | 22.1 | 58.0 | 21.9 | 34.0 | 30.8 | 52.7 | 36.4 | 40.0 | 22.8 | 47.7 | 11.2 | 27.2 |
| Med-Flamingo | 27.4 | 61.9 | 12.7 | 34.0 | 11.8 | 40.2 | 21.1 | 24.4 | 24.3 | 50.4 | 2.4 | 25.7 |
| RadFM | 30.5 | 64.1 | 41.6 | 45.4 | 38.6 | 61.0 | 44.2 | 47.9 | 24.8 | 51.4 | 10.1 | 29.8 |
| LLaVA-Med-7B | 21.6 | 40.5 | 28.2 | 30.1 | 37.0 | 58.4 | 39.2 | 44.9 | 28.5 | 60.1 | 12.3 | 33.6 |
| HuatuoGPT-V-7B | 49.7 | 75.0 | 50.7 | 58.5 | 55.0 | 78.9 | 55.6 | 63.2 | 34.2 | 65.8 | **36.5** | 45.5 |
| VGRefine-7B (Ours) | **51.2** | **76.3** | **52.3** | **59.9** | **56.5** | **80.0** | **56.7** | **64.4** | **36.1** | **68.1** | **36.5** | **46.9** |

## F    COMPARISON WITH OTHER ATTENTION-BASED METHODS

We conducted an additional experiment comparing VGRefine with three very recent attention-based methods for medical MLLMs. Specifically, PAI (Liu et al., 2024c) and AdaptVis (Chen et al., 2025b) aim to refine/manipulate attention maps over visual tokens, while ViCrop (Zhang et al., 2025a) uses attention maps to enhance visual perception.

For a fair comparison, we implemented all methods on HuatuoGPT-V-7B, following their official code and hyperparameter settings. The experimental results, shown in Tab. F.4, indicate that VGRefine consistently outperforms all other methods.

Table F.4: Accuracy on closed-ended medical VQA datasets.

| Model | VQA-RAD | SLAKE | PathVQA | PMC-VQA | Avg. |
|---|---|---|---|---|---|
| HuatuoGPT-V-7B (Baseline) | 67.4 | 76.5 | 60.7 | 53.9 | 65.3 |
| PAI(Liu et al., 2024c) | 43.7 | 24.48 | 20.8 | 52.8 | 33.3 |
| AdaptVis(Chen et al., 2025b) | 68.6 | 75.1 | **67.6** | 52.9 | 66.7 |
| ViCrop (Zhang et al., 2025a) | 68.9 | 70.9 | 66.7 | 54.6 | 65.5 |
| **VGRefine (Ours)** | **71.2** | **76.9** | **67.6** | **56.2** | **68.4** |

## G    MORE EXPERIMENTS ON LARGER MODELS

In this section, we provide more experimental results on larger models (with parameters > 10B). We show comparison on all six benchmarks that are designed for biomedical MLLM evaluation, including VQA-RAD (Lau et al., 2018), SLAKE (Liu et al., 2021a), PathVQA (He et al., 2020), PMC-VQA (Zhang et al., 2023b), OmniMedVQA (Hu et al., 2024) (open-access split), and MMMU (Health & Medicine track) (Yue et al., 2024). All evaluations were conducted in a zero-shot setting using question templates provided by LLaVA (see Sec. I).

*All experiments are conducted using the same hyperparameters across benchmarks.* Specifically, for Step I, we aggregate the attention maps from the top $K = 20$ heads with the highest alignment to visual relevant regions, as measured by KL divergence on our curated evaluation set built using COCO images. This setup prevents data leakage from medical evaluation benchmarks and demonstrates that our method generalizes from natural images to biomedical domains. Low-activation regions are suppressed based on a percentile threshold $p = 50\%$ over attention magnitude. For Step II we apply the attention knockout only at $\ell = 34, 35, 36$ layer, which, according to our analysis in Fig. 3 demonstrates the most relevancy to visual grounding among all the layers. We applied our inference-time method VGRefine-34B on HuatuoGPT-V-34B (Chen et al., 2024a). The hyperparameters $K$ and $p$ are kept consistent with the VGRefine-7B setting. In Step II, we apply attention knockout to more layers, as the 34B model has twice as many layers as the 7B variant and requires deeper intervention to achieve significant improvements.

Results in Tab. G.5 demonstrate that our proposed method consistently achieves good performance across all 6 benchmarks, demonstrating its effectiveness in enhancing all types of medical VQA.

Table G.5: Experiment results of larger models (more than 10B parameters). We evaluate all models under the zero-shot setting. Our inference-time method VGRefine outperforms other state-of-the-art medical MLLMs in most cases. These results underscore that enhanced visual grounding contributes to better performance on medical VQA tasks. It is important to note that VILA-M3 (Nath et al., 2024), MedRegA (Wang et al., 2025) incorporate training data from VQA-RAD (Lau et al., 2018), SLAKE (Liu et al., 2021a), PathVQA (He et al., 2020), and PMC-VQA (Zhang et al., 2023b), thus making zero-shot evaluation unfair, and are excluded from our zero-shot comparison of these benchmarks.

| Benchmarks | Subset | Metric | LLaVA-v1.6-34B | VILA-M3-13B | MedRegA-34B | HuatuoGPT-V-34B | VGRefine-34B (Ours) |
|---|---|---|---|---|---|---|---|
| VQA-RAD | - | CloseAcc | 58.6 | - | - | 68.1 | **72.9** |
| | | BLEU-1 | 44.5 | - | - | 50.5 | **52.6** |
| | | BERT | 69.2 | - | - | **74.8** | **74.8** |
| | | OpenRecall | 43.6 | - | - | 51.7 | **52.8** |
| SLAKE | - | CloseAcc | 67.3 | - | - | 76.9 | **79.1** |
| | | BLEU-1 | 48.6 | - | - | 56.3 | **57.2** |
| | | BERT | 51.8 | - | - | 77.6 | **79.5** |
| | | OpenRecall | 54.2 | - | - | 57.5 | **58.5** |
| PathVQA | - | CloseAcc | 59.1 | - | - | 63.5 | **69.7** |
| | | BLEU-1 | 28.1 | - | - | 36.6 | **37.6** |
| | | BERT | 57.7 | - | - | 65.6 | **65.9** |
| | | OpenRecall | 29.3 | - | - | 36.9 | **37.1** |
| PathVQA | - | CloseAcc | 44.4 | - | - | 58.2 | **58.7** |
| Avg. on Med-VQAs | - | CloseAcc | 57.4 | - | - | 67.0 | **70.7** |
| MMMU | BMS | CloseAcc | 56.4 | 36.8 | 54.3 | 64.3 | **66.0** |
| | CM | | 52.8 | 38.8 | 53.5 | 56.5 | **58.2** |
| | DLM | | 42.6 | 29.0 | 37.7 | 45.1 | **45.4** |
| | P | | 41.6 | 29.3 | 38.4 | 43.7 | **44.0** |
| | PH | | 38.4 | 32.2 | 40.7 | 43.8 | **44.8** |
| | Avg. | | 45.6 | 33.3 | 44.7 | 50.1 | **51.3** |
| OmniMedVQA | CT | CloseAcc | 50.6 | 56.9 | 62.5 | 69.7 | **71.7** |
| | FP | | 63.4 | 50.1 | 80.4 | **84.6** | 84.4 |
| | MRI | | 60.9 | 52.9 | 72.7 | 69.7 | **73.9** |
| | OCT | | 68.4 | 41.5 | 86.2 | **87.8** | 87.6 |
| | Der | | 65.7 | 45.1 | **79.9** | 70.2 | 70.9 |
| | Mic | | 62.8 | 50.6 | 71.3 | 71.1 | **71.4** |
| | X-Ray | | 74.7 | 62.5 | 78.7 | 83.8 | **84.7** |
| | US | | 44.5 | 47.1 | 49.4 | 81.7 | **83.1** |
| | Avg. | | 61.4 | 52.3 | 70.3 | 74.4 | **76.6** |

## H HuatuoGPT-Vision-Bio with BiomedCLIP Vision Encoder

**Model Setup.** To evaluate the effect of domain-specific visual encoders, we modified the original HuatuoGPT-Vision architecture by replacing its CLIP-based vision encoder with BioMed-CLIP Zhang et al. (2025b), a biomedical foundation model pretrained on 15 million scientific image–text pairs. All other components of the model (including the Qwen2 language model, the cross-modal connector module, and the training protocol) remain identical to the original configuration. *This substitution allows us to isolate the impact of specialized medical image representations on visual grounding performance.*

**Training Details.** Since the original training code for HuatuoGPT-Vision was not publicly available, we replicated the training pipeline using the LLaVA-NeXT Liu et al. (2024b) codebase. We follow a two-stage training protocol on the same pretraining and instruction-tuning datasets used in HuatuoGPT-Vision, including LLaVA and PubMedVision. In Stage I, we freeze both the BioMed-CLIP vision encoder and the Qwen2 language model, training only the connector to align visual and textual representations. In Stage II, we fine-tune both the connector and the language model while keeping the vision encoder frozen. The model is trained for 1 epoch. BioMedCLIP processes images at a fixed resolution of $224 \times 224$ with a patch size of 16, which differs from the resolution and tokenization settings used in the original CLIP-based HuatuoGPT-Vision.

**Analysis Results.** As shown in main paper Fig. 1 and Fig. 3, *the issue of suboptimal visual grounding on medical images cannot be solved by using BiomedCLIP vision encoder.*

## I Prompts for VGMED and QA evaluation

### I.1 Prompts for constructing VGMED

---

**Localization Question Set**

```
• Is there a {label} in the image?
• Can you see a {label} in the image?
• Does the image contain a {label}?
• Is a {label} present in this image?
• Do you see a {label} in the picture?
• Is the {label} visible in the image?
• Is there any sign of a {label} in the image?
• Can a {label} be found in this image?
• Does this image show a {label}?
• Is a {label} shown in the picture?
```

---

Figure I.2: The question from a predefined question set is sampled for generating localization questions in both the COCO and VGMED datasets. {label} represents the object (in COCO) or organ/lesion (in VGMED) identified by a bounding box in the corresponding image.

### I.2 Prompts for Zero-shot Evaluation

We used the LLaVA prompt template during the evaluation for open, closed-ended, and multiple-choice questions.

---

**Prompt for VGMED Attribute Questions (MRI)**

Your task is to generate clinically meaningful questions for an evaluation dataset to assess visual grounding capability in medical reasoning, without requiring deep semantic grounding.

- Semantic Grounding:  Anchors linguistic representations to domain-specific medical knowledge to inform what to look for, ensuring accurate reasoning about diseases or anatomical concepts.
- Visual Grounding:  Localizes and interprets specific regions in medical images based on relevant features, enabling spatial alignment of language queries to visual elements for accurate analysis.

We will provide the label of a medical structure (organ, lesion, or tissue) in a {modality} image, derived from a bounding box annotation.  Your need to generate three clinically relevant questions about visual attributes of this structure.

**Guidelines for the question:**

- Focus on visual grounding, without requiring deep medical semantic grounding.
- Ensure clinical relevance.
- Require attention to the entire annotated bounding box.
- Address only observable visual characteristics (e.g., size, shape, density, enhancement, homogeneity).
- Avoid referencing other body parts or surrounding structures.
- Do not include position, modality, or plane.
- Exclude diagnoses or treatments requiring deep semantic grounding.
- Avoid compound or multi-condition questions.
- Ensure variety across the three questions.

**Example questions:**

- "Is the lesion hyper or hypointense?"
- "Is the lesion enhancing?"
- "What does the area of necrosis look like?"
- "What pattern of enhancement does the lesion show?"

Now, generate three questions based on the label.  Return exactly three questions without any additional text or formatting.

**Label:** {label}

---

Figure I.3: For attribute questions in VGMED, we use a specific prompt for each modality. In the prompt, {modality} denotes the modality of the image. {label} denotes the organ or lesion labeled by a bounding box in the image.

---

**Prompt for VGMED Attribute Questions (CT)**

Your task is to generate clinically meaningful questions for an evaluation dataset to assess visual grounding capability in medical reasoning, without requiring deep semantic grounding.

- Semantic Grounding: Anchors linguistic representations to domain-specific medical knowledge to inform what to look for, ensuring accurate reasoning about diseases or anatomical concepts.

- Visual Grounding: Localizes and interprets specific regions in medical images based on relevant features, enabling spatial alignment of language queries to visual elements for accurate analysis.

We will provide the label of a medical structure (organ, lesion, or tissue) in a {modality} image, derived from a bounding box annotation. Your need to generate three clinically relevant questions about visual attributes of this structure.

**Guidelines for the question:**

- Focus on visual grounding, without requiring deep medical semantic grounding.

- Ensure clinical relevance.

- Require attention to the entire annotated bounding box.

- Address only observable visual characteristics (e.g., size, shape, density, enhancement, homogeneity).

- Avoid referencing other body parts or surrounding structures.

- Do not include position, modality, or plane.

- Exclude diagnoses or treatments requiring deep semantic grounding.

- Avoid compound or multi-condition questions.

- Ensure variety across the three questions.

**Example questions:**

- "Are there ground glass opacities within the lung?"

- "Is the kidney enlarged?"

- "What is the size of the necrosis?"

Now, generate three questions based on the label. Return exactly three questions without any additional text or formatting.

**Label:** {label}

Figure I.4: For attribute questions in VGMED, we use a specific prompt for each modality. In the prompt, {modality} denotes the modality of the image. {label} denotes the organ or lesion labeled by a bounding box in the image.

---

**Prompt for VGMED Attribute Questions (Ultrasound)**

Your task is to generate clinically meaningful questions for
an evaluation dataset to assess visual grounding capability
in medical reasoning, without requiring deep semantic
grounding.

- Semantic Grounding:  Anchors linguistic representations
  to domain-specific medical knowledge to inform what to
  look for, ensuring accurate reasoning about diseases or
  anatomical concepts.
- Visual Grounding:  Localizes and interprets specific
  regions in medical images based on relevant features,
  enabling spatial alignment of language queries to visual
  elements for accurate analysis.

We will provide the label of a medical structure (organ,
lesion, or tissue) in a {modality} image, derived from
a bounding box annotation.  Your need to generate three
clinically relevant questions about visual attributes of this
structure.
**Guidelines for the question**:

- Focus on visual grounding, without requiring deep medical
  semantic grounding.
- Ensure clinical relevance.
- Require attention to the entire annotated bounding box.
- Address only observable visual characteristics (e.g.,
  size, shape, density, enhancement, homogeneity).
- Avoid referencing other body parts or surrounding
  structures.
- Do not include position, modality, or plane.
- Exclude diagnoses or treatments requiring deep semantic
  grounding.
- Avoid compound or multi-condition questions.
- Ensure variety across the three questions.

**Example questions**:

- "Does the thyroid nodule have irregular or microlobulated
  margins?"
- "Does the thyroid nodule have marked hypoechogenicity?"
- "Does the thyroid nodule have multiple
  microcalcifications?"
- "Is the breast lesion homogeneous or heterogeneous?"
- "Does the breast lesion appear solid or cystic on
  ultrasound?"

Now, generate three questions based on the label.  Return
exactly three questions without any additional text or
formatting.

**Label:**  {label}

---

Figure I.5: For attribute questions in VGMED, we use a specific prompt for each modality. In the prompt, {modality} denotes the modality of the image. {label} denotes the organ or lesion labeled by a bounding box in the image.

---

**Prompt for VGMED Attribute Questions (X-ray)**

```
Your task is to generate clinically meaningful questions for
an evaluation dataset to assess visual grounding capability
in medical reasoning, without requiring deep semantic
grounding.
```
- ```
  Semantic Grounding:  Anchors linguistic representations
  to domain-specific medical knowledge to inform what to
  look for, ensuring accurate reasoning about diseases or
  anatomical concepts.
  ```
- ```
  Visual Grounding:  Localizes and interprets specific
  regions in medical images based on relevant features,
  enabling spatial alignment of language queries to visual
  elements for accurate analysis.
  ```

```
We will provide the label of a medical structure (organ,
lesion, or tissue) in a {modality} image, derived from
a bounding box annotation.  Your need to generate three
clinically relevant questions about visual attributes of this
structure.
```
**Guidelines for the question:**
- ```
  Focus on visual grounding, without requiring deep medical
  semantic grounding.
  ```
- `Ensure clinical relevance.`
- `Require attention to the entire annotated bounding box.`
- ```
  Address only observable visual characteristics (e.g.,
  size, shape, density, enhancement, homogeneity).
  ```
- ```
  Avoid referencing other body parts or surrounding
  structures.
  ```
- `Do not include position, modality, or plane.`
- ```
  Exclude diagnoses or treatments requiring deep semantic
  grounding.
  ```
- `Avoid compound or multi-condition questions.`
- `Ensure variety across the three questions.`

**Example questions:**
- `"What is the size of the pneumothorax?"`
- `"Where is the pneumothorax?"`
- ```
  "Does the lung field appear more opaque or translucent in
  the annotated region?"
  ```

```
Now, generate three questions based on the label.  Return
exactly three questions without any additional text or
formatting.
```

**Label:** `{label}`

Figure I.6: For attribute questions in VGMED, we use a specific prompt for each modality. In the prompt, {modality} denotes the modality of the image. {label} denotes the organ or lesion labeled by a bounding box in the image.

---

**Prompt for VGMED Attribute Questions (Fundus Photography)**

Your task is to generate clinically meaningful questions for an evaluation dataset to assess visual grounding capability in medical reasoning, without requiring deep semantic grounding.

- Semantic Grounding: Anchors linguistic representations to domain-specific medical knowledge to inform what to look for, ensuring accurate reasoning about diseases or anatomical concepts.

- Visual Grounding: Localizes and interprets specific regions in medical images based on relevant features, enabling spatial alignment of language queries to visual elements for accurate analysis.

We will provide the label of a medical structure (organ, lesion, or tissue) in a {modality} image, derived from a bounding box annotation. Your need to generate three clinically relevant questions about visual attributes of this structure.

**Guidelines for the question:**

- Focus on visual grounding, without requiring deep medical semantic grounding.

- Ensure clinical relevance.

- Require attention to the entire annotated bounding box.

- Address only observable visual characteristics (e.g., size, shape, density, enhancement, homogeneity).

- Avoid referencing other body parts or surrounding structures.

- Do not include position, modality, or plane.

- Exclude diagnoses or treatments requiring deep semantic grounding.

- Avoid compound or multi-condition questions.

- Ensure variety across the three questions.

**Example questions:**

- "Is there any pallor observed in the optic disc?"

- "Does the optic disc appear swollen or elevated?"

- "Is there any evidence of swelling or pallor in the optic disc?"

Now, generate three questions based on the label. Return exactly three questions without any additional text or formatting.

**Label:** {label}

---

Figure I.7: For attribute questions in VGMED, we use a specific prompt for each modality. In the prompt, {modality} denotes the modality of the image. {label} denotes the organ or lesion labeled by a bounding box in the image.

---

**Prompt for VGMED Attribute Questions (Endoscopy)**

```
Your task is to generate clinically meaningful questions for
an evaluation dataset to assess visual grounding capability
in medical reasoning, without requiring deep semantic
grounding.
```

- ```
  Semantic Grounding:  Anchors linguistic representations
  to domain-specific medical knowledge to inform what to
  look for, ensuring accurate reasoning about diseases or
  anatomical concepts.
  ```
- ```
  Visual Grounding:  Localizes and interprets specific
  regions in medical images based on relevant features,
  enabling spatial alignment of language queries to visual
  elements for accurate analysis.
  ```

```
We will provide the label of a medical structure (organ,
lesion, or tissue) in a {modality} image, derived from
a bounding box annotation.  Your need to generate three
clinically relevant questions about visual attributes of this
structure.
```
**Guidelines for the question:**

- ```
  Focus on visual grounding, without requiring deep medical
  semantic grounding.
  ```
- ```
  Ensure clinical relevance.
  ```
- ```
  Require attention to the entire annotated bounding box.
  ```
- ```
  Address only observable visual characteristics (e.g.,
  size, shape, density, enhancement, homogeneity).
  ```
- ```
  Avoid referencing other body parts or surrounding
  structures.
  ```
- ```
  Do not include position, modality, or plane.
  ```
- ```
  Exclude diagnoses or treatments requiring deep semantic
  grounding.
  ```
- ```
  Avoid compound or multi-condition questions.
  ```
- ```
  Ensure variety across the three questions.
  ```

**Example questions:**

- ```
  "What is the size of the polyp?"
  ```
- ```
  "Does the colorectal polyp have a smooth or lobulated
  surface appearance?"
  ```
- ```
  "What is the mobility of the polyp?"
  ```

```
Now, generate three questions based on the label.  Return
exactly three questions without any additional text or
formatting.
```

**Label:** {label}

---

Figure I.8: For attribute questions in VGMED, we use a specific prompt for each modality. In the prompt, {modality} denotes the modality of the image. {label} denotes the organ or lesion labeled by a bounding box in the image.

---

**Prompt for VGMED Attribute Questions (PET)**

```
Your task is to generate clinically meaningful questions for
an evaluation dataset to assess visual grounding capability
in medical reasoning, without requiring deep semantic
grounding.
```

- `Semantic Grounding:  Anchors linguistic representations`
  `to domain-specific medical knowledge to inform what to`
  `look for, ensuring accurate reasoning about diseases or`
  `anatomical concepts.`

- `Visual Grounding:  Localizes and interprets specific`
  `regions in medical images based on relevant features,`
  `enabling spatial alignment of language queries to visual`
  `elements for accurate analysis.`

```
We will provide the label of a medical structure (organ,
lesion, or tissue) in a {modality} image, derived from
a bounding box annotation.  Your need to generate three
clinically relevant questions about visual attributes of this
structure.
```
**Guidelines for the question:**

- `Focus on visual grounding, without requiring deep medical`
  `semantic grounding.`

- `Ensure clinical relevance.`

- `Require attention to the entire annotated bounding box.`

- `Address only observable visual characteristics (e.g.,`
  `size, shape, density, enhancement, homogeneity).`

- `Avoid referencing other body parts or surrounding`
  `structures.`

- `Do not include position, modality, or plane.`

- `Exclude diagnoses or treatments requiring deep semantic`
  `grounding.`

- `Avoid compound or multi-condition questions.`

- `Ensure variety across the three questions.`

**Example questions:**

- `"Does the lesion show increased radiotracer uptake on the`
  `PET scan?"`

- `"Is the lesion hypo- or hyper-metabolic?"`

- `"What is the Standardized Uptake Value (SUV) of the`
  `lesion?"`

```
Now, generate three questions based on the label.  Return
exactly three questions without any additional text or
formatting.
```

**Label:**  `{label}`

Figure I.9: For attribute questions in VGMED, we use a specific prompt for each modality. In the prompt, {modality} denotes the modality of the image. {label} denotes the organ or lesion labeled by a bounding box in the image.

---

**Prompt for VGMED Attribute Questions (Dermoscopy)**

Your task is to generate clinically meaningful questions for an evaluation dataset to assess visual grounding capability in medical reasoning, without requiring deep semantic grounding.

- Semantic Grounding: Anchors linguistic representations to domain-specific medical knowledge to inform what to look for, ensuring accurate reasoning about diseases or anatomical concepts.
- Visual Grounding: Localizes and interprets specific regions in medical images based on relevant features, enabling spatial alignment of language queries to visual elements for accurate analysis.

We will provide the label of a medical structure (organ, lesion, or tissue) in a {modality} image, derived from a bounding box annotation. Your need to generate three clinically relevant questions about visual attributes of this structure.

**Guidelines for the question:**

- Focus on visual grounding, without requiring deep medical semantic grounding.
- Ensure clinical relevance.
- Require attention to the entire annotated bounding box.
- Address only observable visual characteristics (e.g., size, shape, density, enhancement, homogeneity).
- Avoid referencing other body parts or surrounding structures.
- Do not include position, modality, or plane.
- Exclude diagnoses or treatments requiring deep semantic grounding.
- Avoid compound or multi-condition questions.
- Ensure variety across the three questions.

**Example questions:**

- "What is the size of the lesion?"
- "Is the lesion hypo- or hyper pigmented?"
- "Does the lesion have peripheral black dots or clods?"
- "Does the skin lesion have thick lines (reticular or branched)?"
- "Does the lesion have Polymorphous vessels?"

Now, generate three questions based on the label. Return exactly three questions without any additional text or formatting.

**Label:** {label}

---

Figure I.10: For attribute questions in VGMED, we use a specific prompt for each modality. In the prompt, {modality} denotes the modality of the image. {label} denotes the organ or lesion labeled by a bounding box in the image.

---

**Prompt for COCO Attribute Questions**

```
Your task is to generate one simple and meaningful question
about a visual attribute of an object identified in an image.

We will provide the label of the object, which comes from a
bounding box annotation in an image from the COCO dataset.
```

**Guidelines for the question:**

- `Ensure variety in the questions generated.`
- `Focus only on the visual characteristics (e.g., color, size, material, etc.)  of the given object.`
- `Do not reference other parts of the image.`
- `Do not ask questions about the position of the object or the surrounding structure.`
- `Avoid compound or multi-condition questions.`

```
Now, generate one question based on the following label:
```
**Label:** `{label}`

---

Figure I.11: In order to compare the results between our VGMED datasets and natural scene images, we have also generated the attribute questions for COCO examples. {label} refers to the object label from the image's bounding boxes.

---

**Short Answer (e.g., VQA-RAD, SLAKE, PathVQA)**

```
<question>
Answer the question using a single word or phrase.
```

---

Figure I.12: Prompt for evaluating the open and closed-ended questions in VQA-RAD, SLAKE, and PathVQA benchmarks.

---

**Option-only for multiple-choice (e.g., PMC-VQA, OmniMedVQA, and MMMU)**

```
<question>
    A. <option_1>
    B. <option_2>
    C. <option_3>
    D. <option_4>
Answer with the option's letter from the given choices
directly.
```

---

Figure I.13: Prompt for evaluating the multiple-choice VQA benchmarks.

# J    ADDITIONAL QUALITATIVE EVALUATION

## J.1    HUMAN EVALUATION

We conducted a blinded human evaluation involving five experienced clinicians (4 of them have over 10 years of clinical practice). The study was based on a 20-case questionnaire. For each case, clinicians were shown a medical image with a VQA question and two corresponding attention maps: (1) from the baseline model and (2) from the same model after applying VGRefine. The source of each attention map was not disclosed, and their order was randomized. Clinicians were asked: "*Which model's attention visualization (shown as heatmap) appears more clinically reasonable and trustworthy?*".

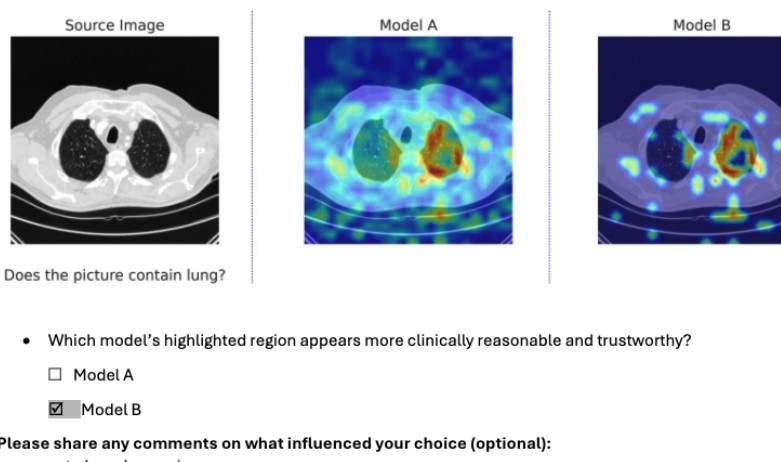

Figure J.14: Example of a blinded human evaluation case, showing a medical image with a VQA question, baseline attention map, and VGRefine attention map, assessed by an experienced clinician for clinical reasonableness. Clinician feedback highlighted that VGRefine attention maps were less noisy, better localized, and more aligned with expected clinical focus points.

## J.2 Additional Qualitative Analysis on Medical MLLM's Attention Maps

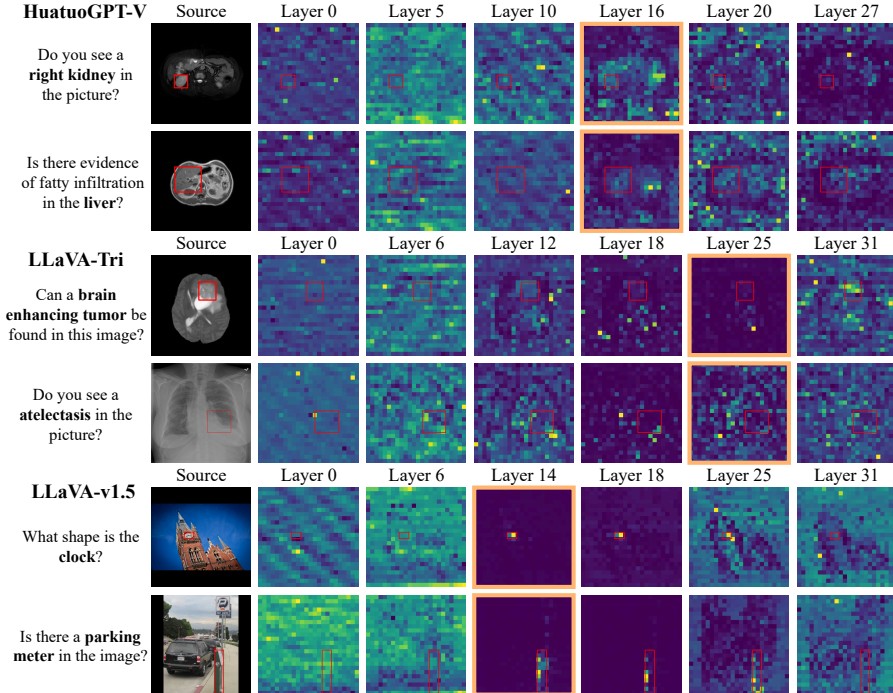

Figure J.15: **Qualitative evaluation.** We visualize attention maps across different layers, including those with the lowest KL divergence (highlighted with an orange boundary), which are indicative of layers most relevant to visual grounding in MLLMs. For medical images, the attention maps of medical MLLMs show limited alignment with the ground-truth annotated regions. In contrast, a general-domain MLLM LLaVA-v1.5 applied to natural images exhibits strong alignment with relevant regions, consistent with other study of general-domain MLLMs (Zhang et al., 2025a). This highlights a gap in MLLM's visual grounding performance between the medical and natural image domains. Best viewed in color and with zoom. **Additional results in Supp J.2**.

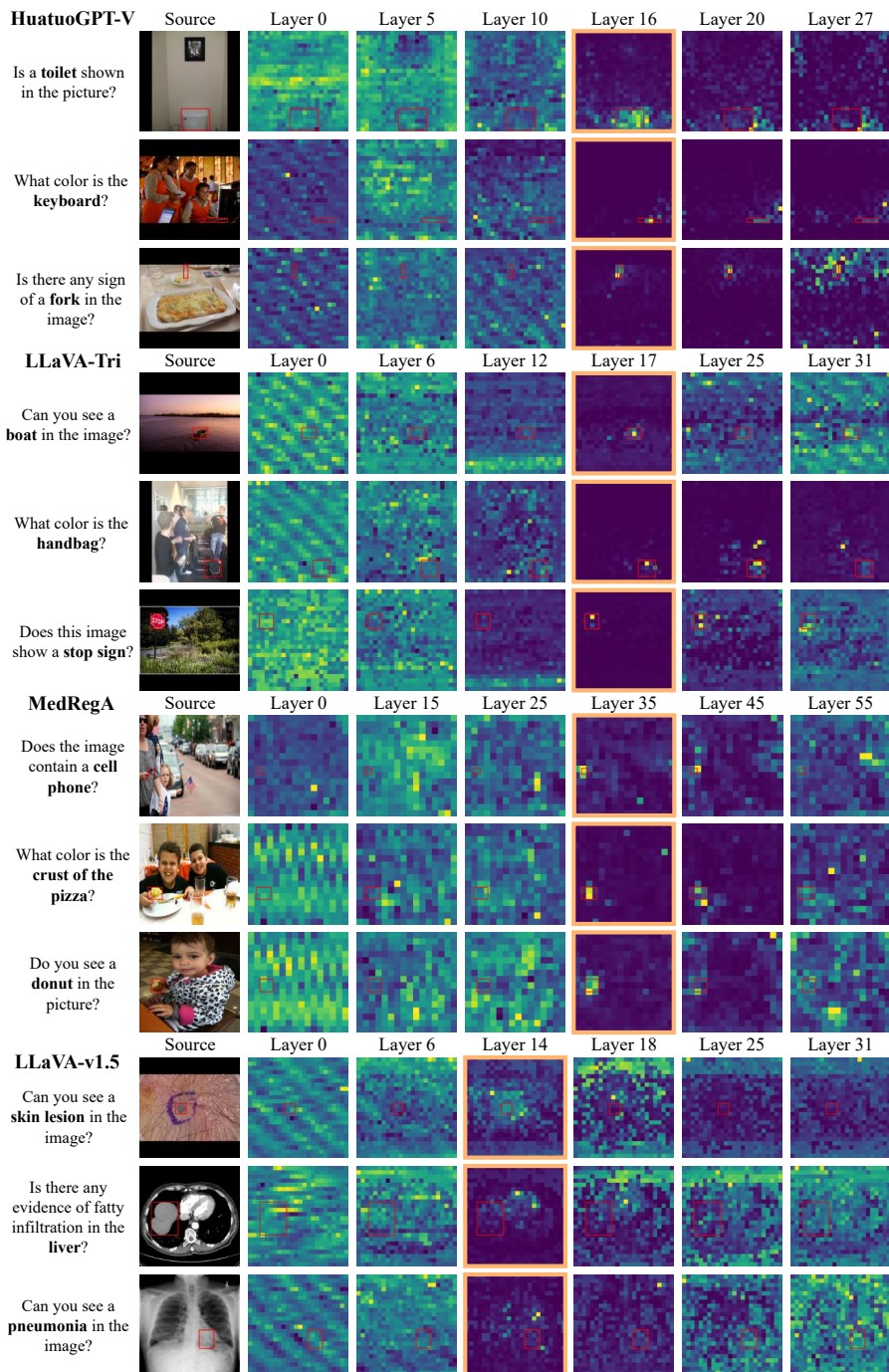

Figure J.16: **Qualitative** evaluation of (i) medical MLLMs HuatuoGPT-V, LLaVA-Tri and MedRegA on COCO, and (ii) LLaVA-v1.5 on VGMED. We visualize attention maps across different layers, including those with the lowest KL divergence (highlighted with an orange boundary), which are indicative of layers most relevant to visual grounding in MLLMs. We observe that LLaVA-v1.5 fails to ground predictions in clinically relevant regions when operating on medical images and medical VQA tasks. Furthermore, medical-domain models can ground their predictions when applied to natural images. This is consistent with our quantitative analysis in Fig. 3 of the main paper. Together, they show that medical MLLMs possess good visual grounding capabilities in general-domain settings. **Overall, this confirms that the grounding failure is not due to model weakness, but is fundamentally specific to the medical domain, consistent with our central findings. Inadequate visual grounding is a medical-domain failure mode.**

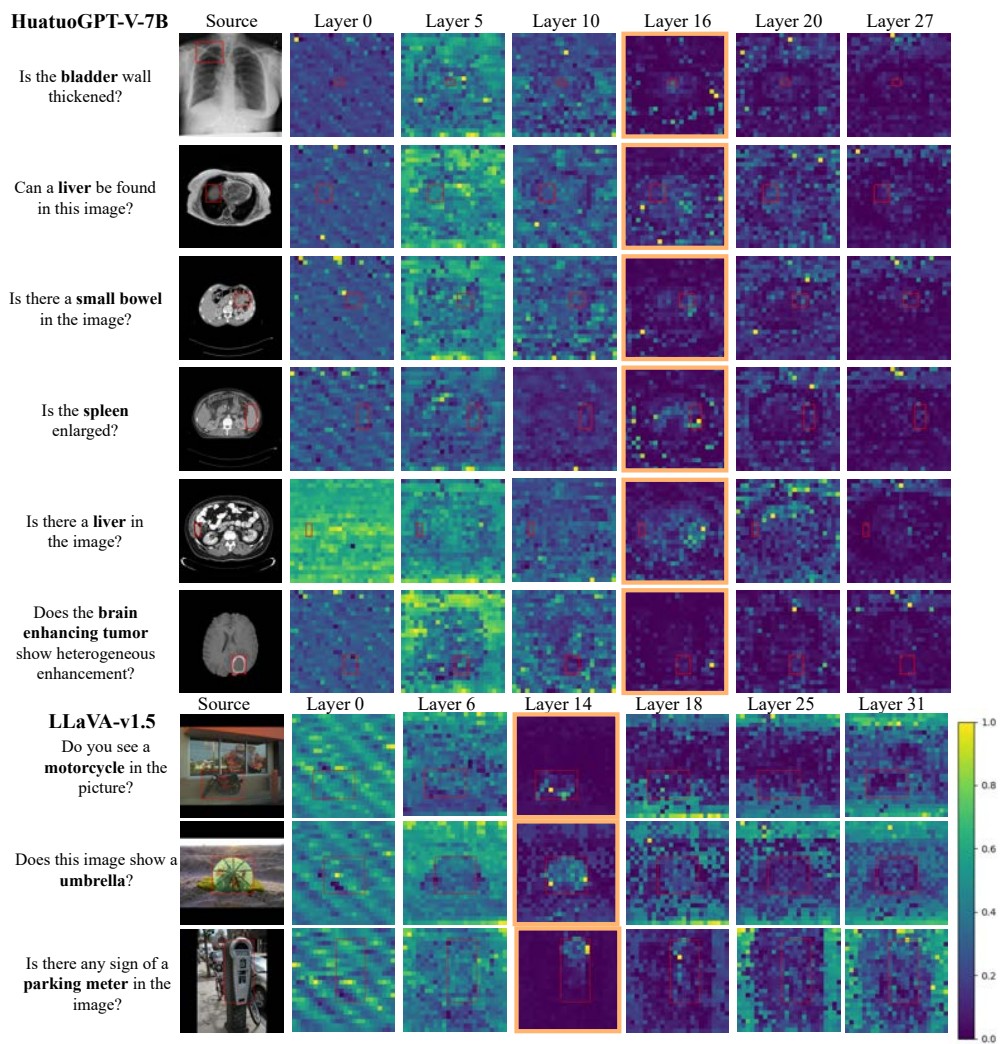

Figure J.17: **Qualitative evaluation.** We visualize attention maps across different layers, including those with the lowest KL divergence (highlighted with an orange boundary), which are indicative of layers most relevant to visual grounding in MLLMs. For medical images, the attention maps of medical MLLMs show limited alignment with the ground-truth annotated regions. In contrast, a general-domain MLLM LLaVA-v1.5 applied to natural images exhibits strong alignment with relevant regions, consistent with other study of general-domain MLLMs Zhang et al. (2025a). This highlights a gap in MLLM's visual grounding performance between the medical and natural image domains.

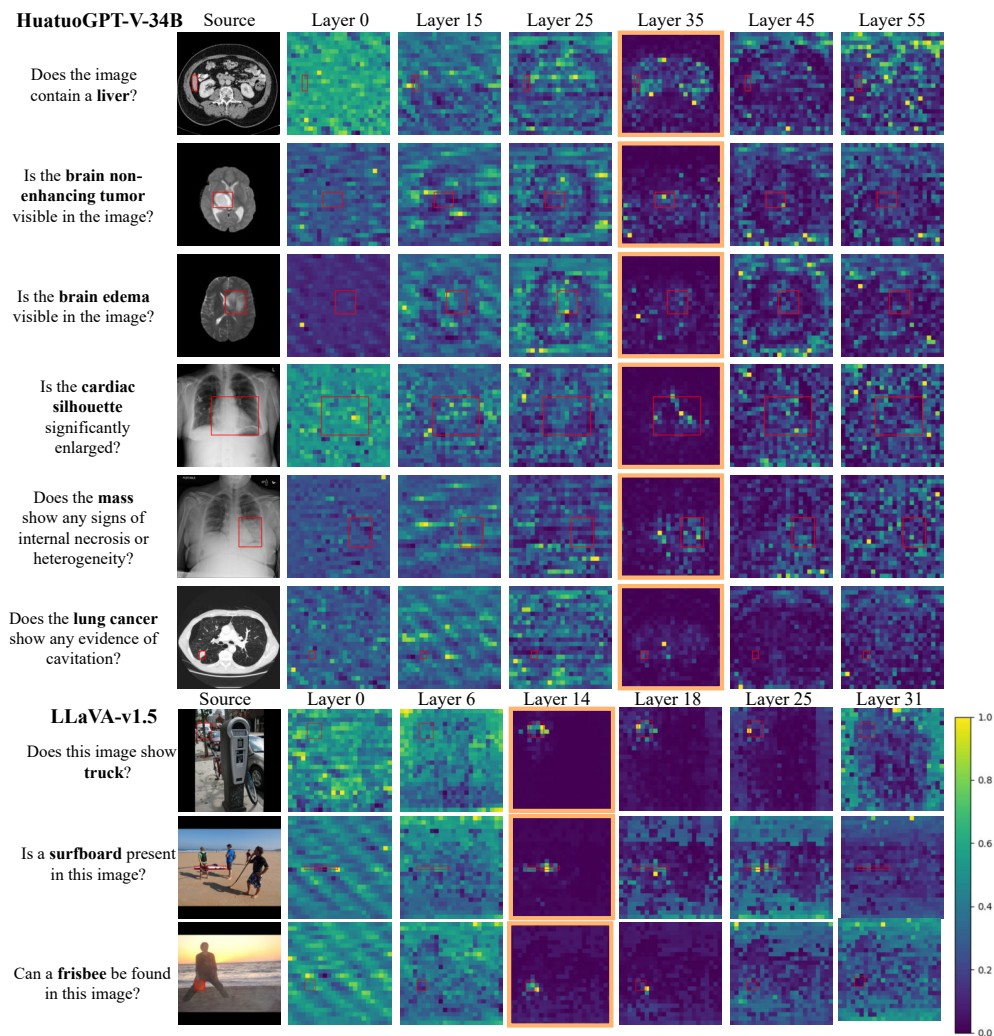

Figure J.18: **Qualitative evaluation.** We visualize attention maps across different layers, including those with the lowest KL divergence (highlighted with an orange boundary), which are indicative of layers most relevant to visual grounding in MLLMs. For medical images, the attention maps of medical MLLMs show limited alignment with the ground-truth annotated regions. In contrast, a general-domain MLLM LLaVA-v1.5 applied to natural images exhibits strong alignment with relevant regions, consistent with other study of general-domain MLLMs Zhang et al. (2025a). This highlights a gap in MLLM's visual grounding performance between the medical and natural image domains.

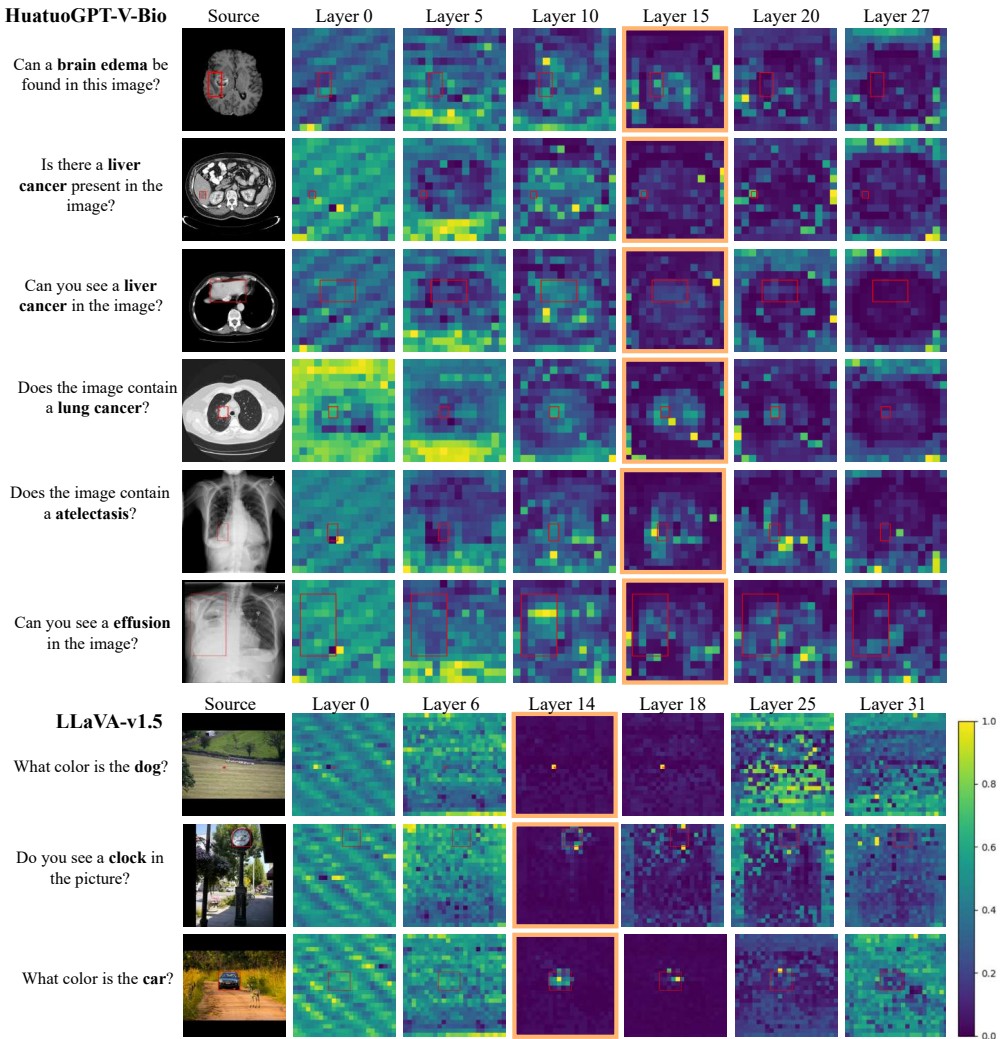

Figure J.19: **Qualitative evaluation.** We visualize attention maps across different layers, including those with the lowest KL divergence (highlighted with an orange boundary), which are indicative of layers most relevant to visual grounding in MLLMs. For medical images, the attention maps of HuatuoGPT-V-Bio with a specialized vision encoder (BiomedCLIP) show limited alignment with the ground-truth annotated regions. In contrast, a general-domain MLLM LLaVA-v1.5 applied to natural images exhibits strong alignment with relevant regions, consistent with other study of general-domain MLLMs Zhang et al. (2025a).

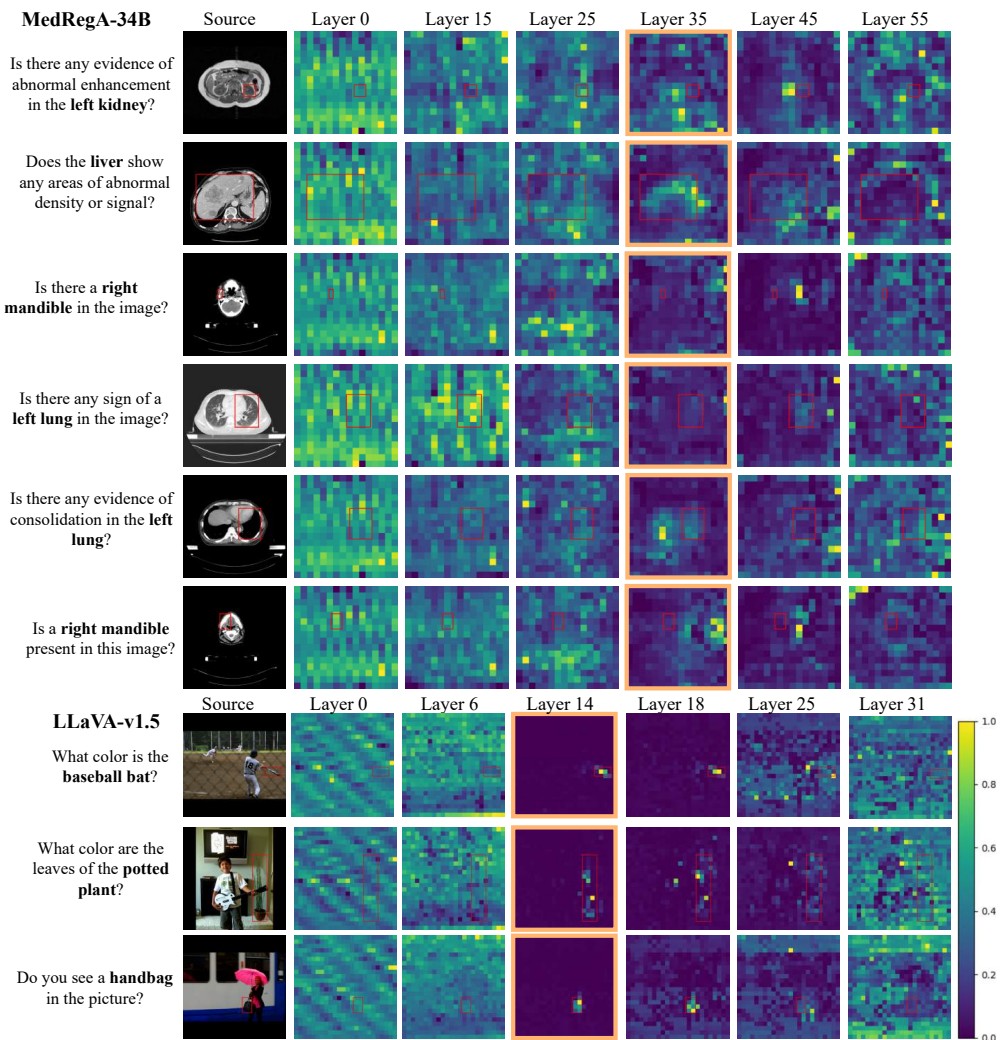

Figure J.20: **Qualitative evaluation.** We visualize attention maps across different layers, including those with the lowest KL divergence (highlighted with an orange boundary), which are indicative of layers most relevant to visual grounding in MLLMs. For medical images, the attention maps of medical MLLMs show limited alignment with the ground-truth annotated regions. In contrast, a general-domain MLLM LLaVA-v1.5 applied to natural images exhibits strong alignment with relevant regions, consistent with other study of general-domain MLLMs Zhang et al. (2025a). This highlights a gap in MLLM's visual grounding performance between the medical and natural image domains.

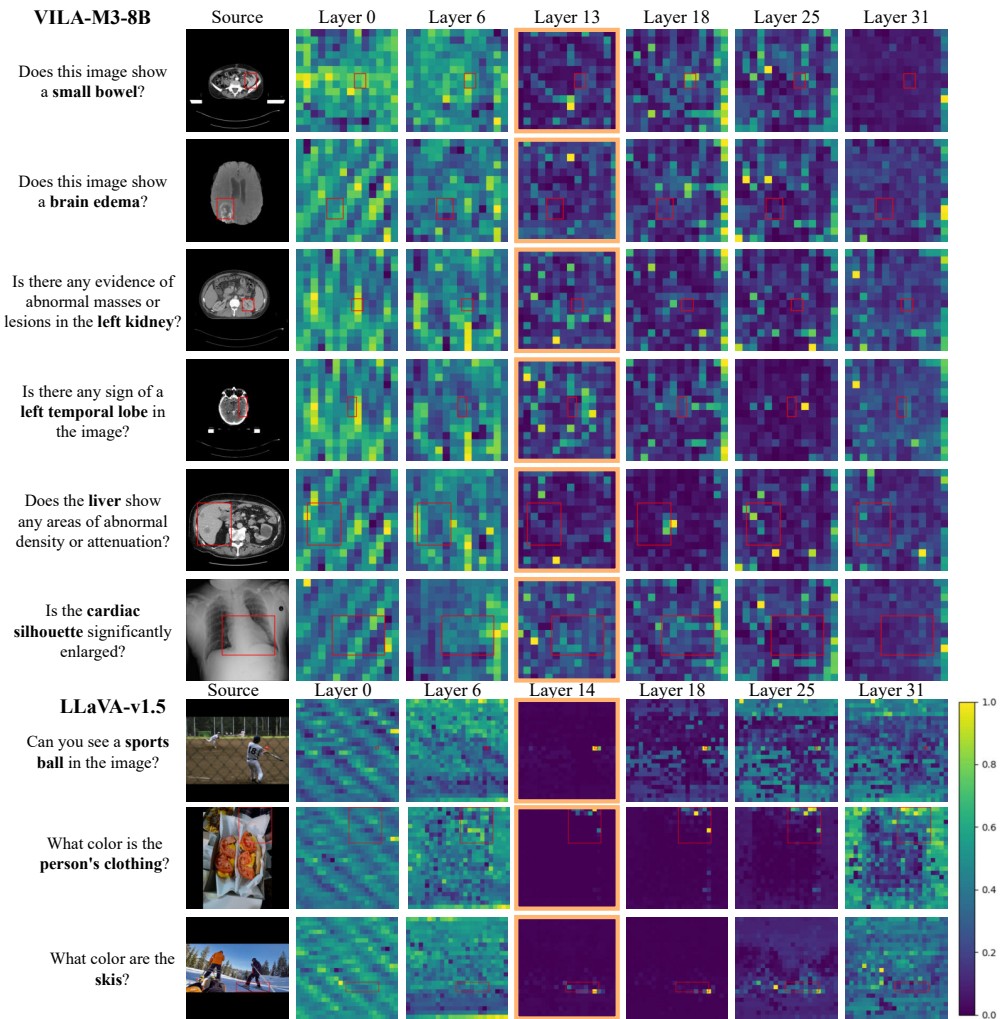

Figure J.21: **Qualitative evaluation.** We visualize attention maps across different layers, including those with the lowest KL divergence (highlighted with an orange boundary), which are indicative of layers most relevant to visual grounding in MLLMs. For medical images, the attention maps of medical MLLMs show limited alignment with the ground-truth annotated regions. In contrast, a general-domain MLLM LLaVA-v1.5 applied to natural images exhibits strong alignment with relevant regions, consistent with other study of general-domain MLLMs Zhang et al. (2025a). This highlights a gap in MLLM's visual grounding performance between the medical and natural image domains.

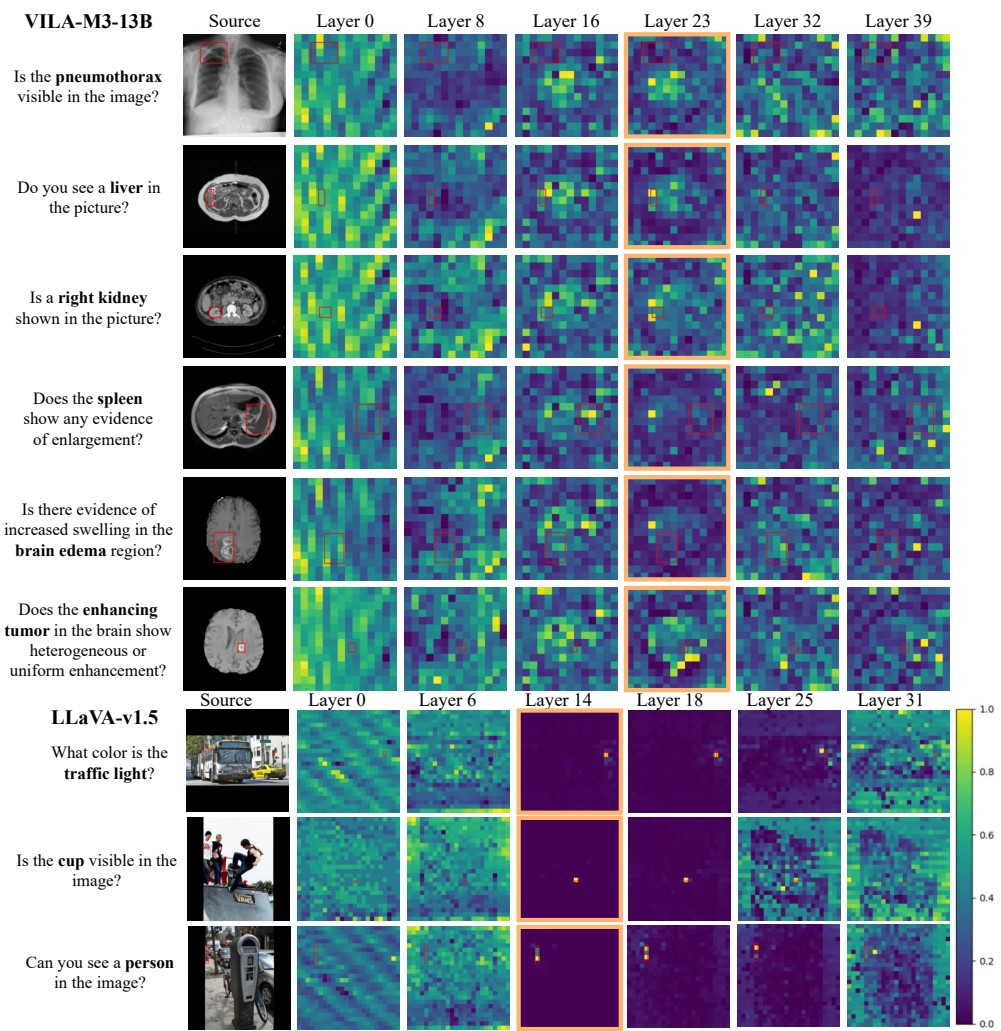

Figure J.22: **Qualitative evaluation.** We visualize attention maps across different layers, including those with the lowest KL divergence (highlighted with an orange boundary), which are indicative of layers most relevant to visual grounding in MLLMs. For medical images, the attention maps of medical MLLMs show limited alignment with the ground-truth annotated regions. In contrast, a general-domain MLLM LLaVA-v1.5 applied to natural images exhibits strong alignment with relevant regions, consistent with other study of general-domain MLLMs Zhang et al. (2025a). This highlights a gap in MLLM's visual grounding performance between the medical and natural image domains.

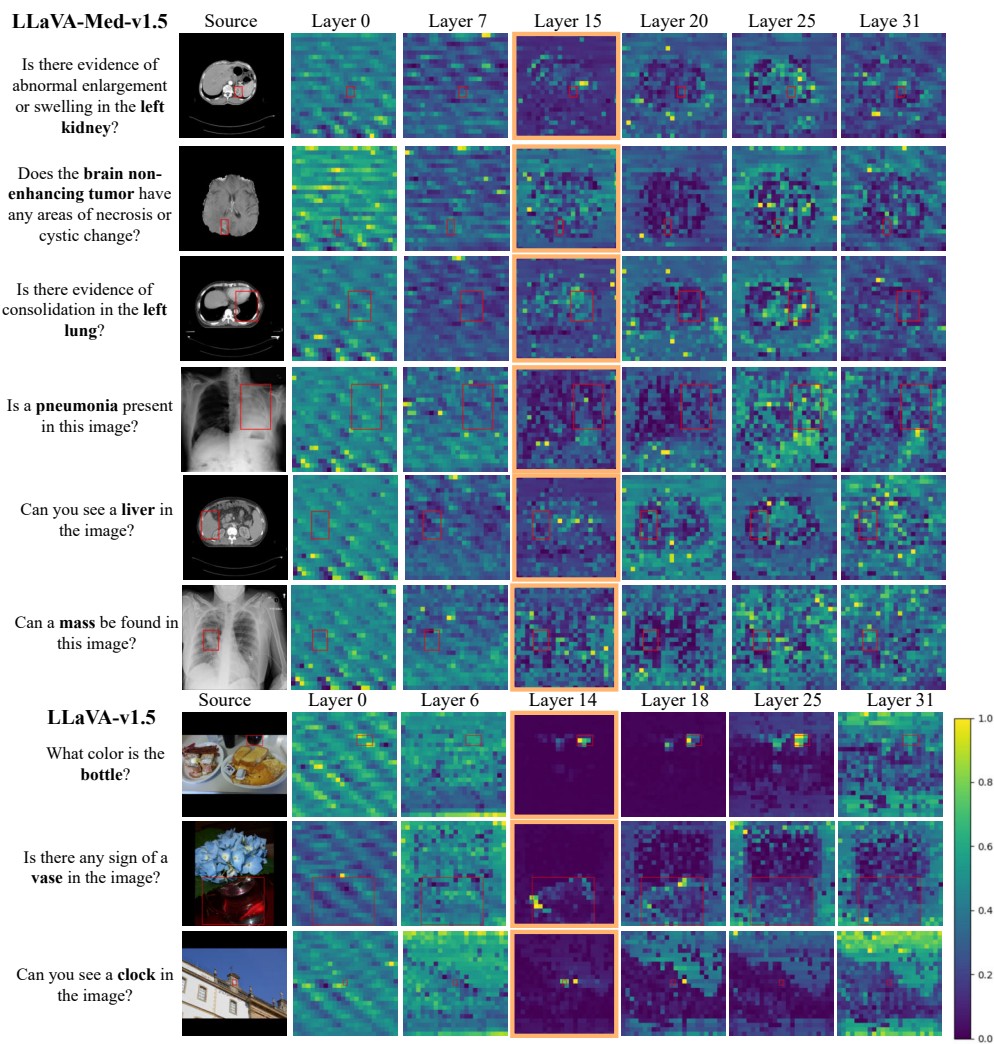

Figure J.23: **Qualitative evaluation.** We visualize attention maps across different layers, including those with the lowest KL divergence (highlighted with an orange boundary), which are indicative of layers most relevant to visual grounding in MLLMs. For medical images, the attention maps of medical MLLMs show limited alignment with the ground-truth annotated regions. In contrast, a general-domain MLLM LLaVA-v1.5 applied to natural images exhibits strong alignment with relevant regions, consistent with other study of general-domain MLLMs Zhang et al. (2025a). This highlights a gap in MLLM's visual grounding performance between the medical and natural image domains.

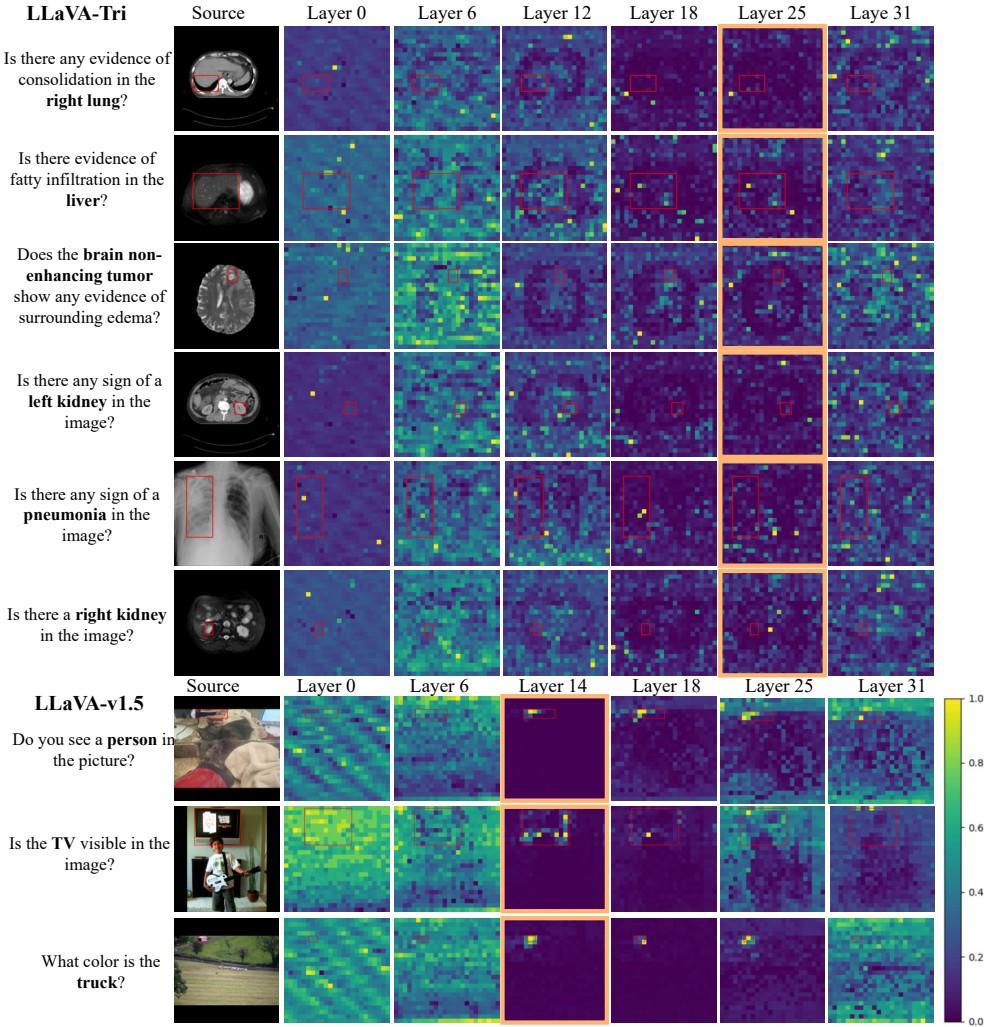

Figure J.24: **Qualitative evaluation.** We visualize attention maps across different layers, including those with the lowest KL divergence (highlighted with an orange boundary), which are indicative of layers most relevant to visual grounding in MLLMs. For medical images, the attention maps of medical MLLMs show limited alignment with the ground-truth annotated regions. In contrast, a general-domain MLLM LLaVA-v1.5 applied to natural images exhibits strong alignment with relevant regions, consistent with other study of general-domain MLLMs Zhang et al. (2025a). This highlights a gap in MLLM's visual grounding performance between the medical and natural image domains.

# K ADDITIONAL EVALUATION OF LATEST GENERAL MLLMS

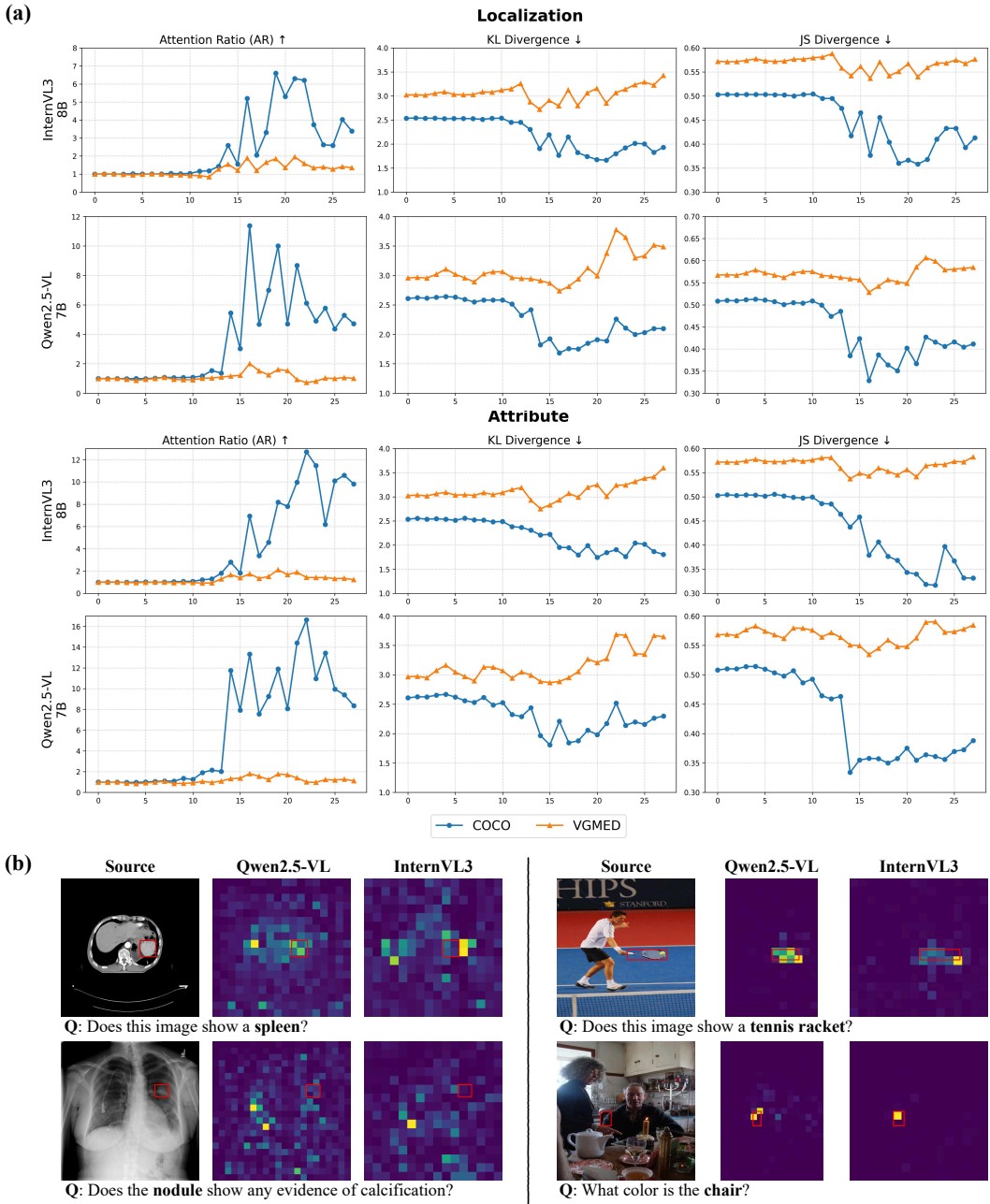

Figure K.25: (a) **Quantitative** and (b) **qualitative** evaluation of InternVL3-8B and Qwen2.5-VL-7B on VGMED and COCO. We observe that the visual grounding deficiency in medical domain persists even in these latest general-purpose models.

# L    ATTENTION MAPS DERIVED FROM DIFFERENT TEXT TOKENS

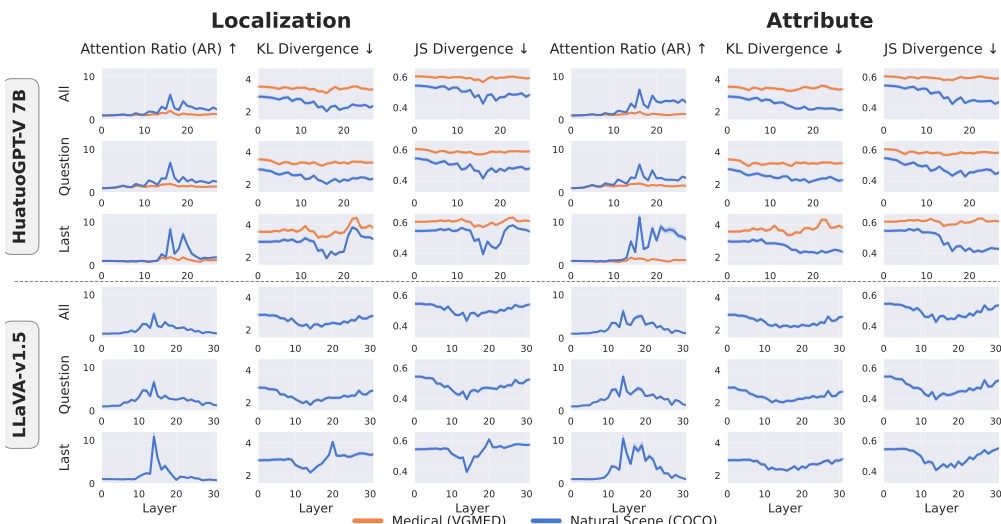

Figure L.26: **Comparison of visual grounding when using *all input tokens*, *question-only tokens*, or the *last token* to derive attention maps.** Using two representative MLLMs (HuatuoGPT-V-7B and LLaVA-v1.5), we evaluate how different token-selection strategies affect attention alignment on VGMED and COCO. Across all metrics and layers, attention maps computed from the *last token* achieves equal or better alignment with ground-truth regions compared to the alternative options.

## M    REPRESENTATIVE FAILURE CASES

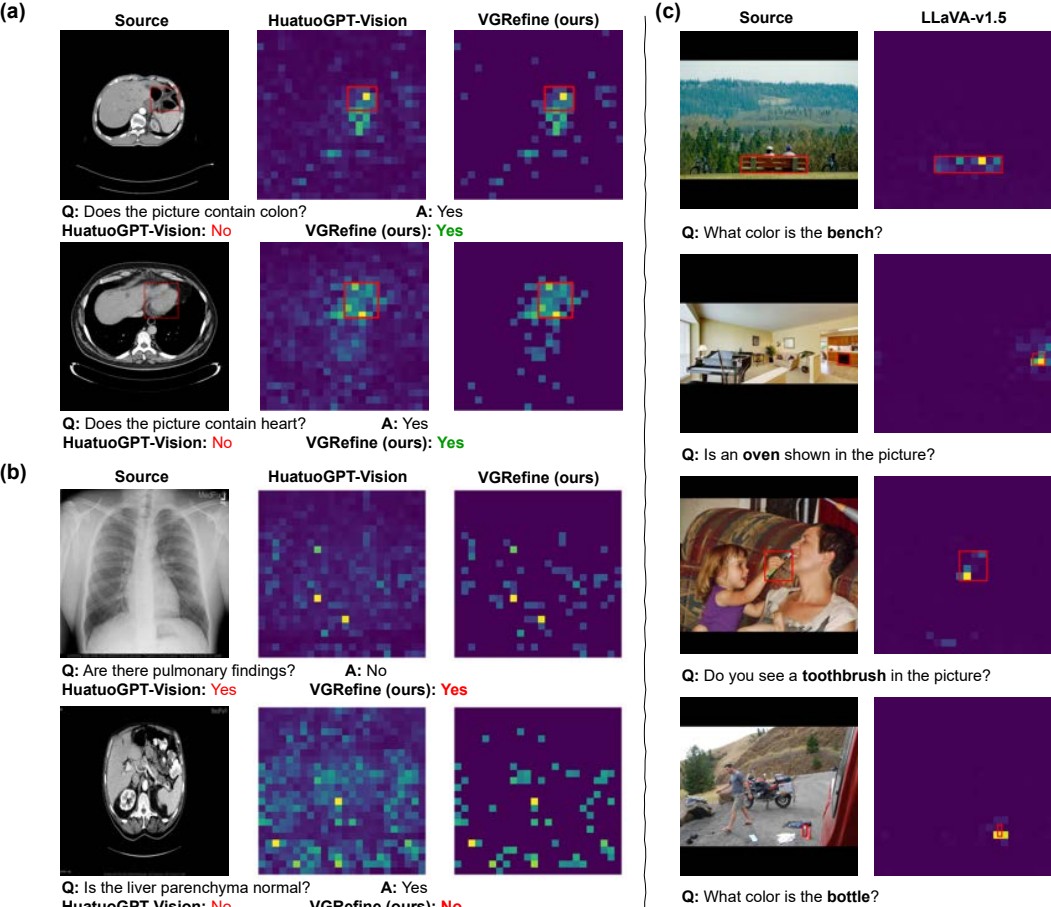

Figure M.27: **Representative failure cases of HuatuoGPT-Vision on medical benchmarks.** (a) The model correctly interprets the question but attends to the wrong anatomical region, leading to an incorrect answer. After applying VGRefine, the model's attention shifts toward more clinically relevant region, resulting in the correct prediction. (b) The model misunderstand the question, resulting in both semantic and visual grounding failure. (c) Additionally, we include examples from LLaVA-v1.5 on natural images as a reference of accurate visual grounding. While multiple factors contribute to poor generalization, weak visual grounding consistently emerges as a major and measurable issue, though not the sole cause.

## N    Clinical Validation During VGMED Curation

As part of the VGMED curation process, clinicians reviewed each sample to verify that (i) the question is properly focused on visual grounding, (ii) it does not require deep or diagnostic-level semantic medical reasoning, and (iii) it remains clinically appropriate and meaningful. An example of the rating interface used during the curation process is shown in Fig. N.28.

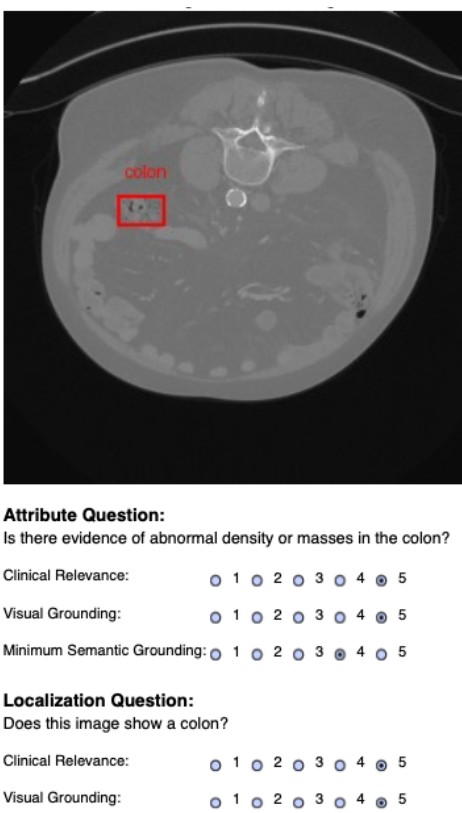

Figure N.28: Example of the clinician rating interface used during VGMED curation.

**Clinical Relevance**

- **1:** Irrelevant or misleading; the question is clinically inappropriate or nonsensical in this context.
- **2:** Marginally relevant; the question has limited medical value or loosely pertains to the case.
- **3:** Acceptable; the question is reasonable in clinical significance.
- **4:** Clinically useful; the question is clearly relevant and meaningful to medical interpretation.
- **5:** Highly relevant and valid; the question is well-phrased, accurate, and directly supports clinical reasoning.

**Visual Grounding**

- **1:** It refers to other anatomy or ignores the boxed area entirely; ignores the region.
- **2:** The question has only a weak or incidental connection to the boxed region; the area is largely irrelevant to the text.
- **3:** It reasonably overlaps or implies the boxed region.

- **4:** Clear reference to the boxed region.
- **5:** Perfectly aligned, the question precisely refers to the boxed region.

**Minimum Semantic Grounding**

- **1:** Very deep semantic grounding; requires advanced, multi-step clinical reasoning, such as staging, prognosis, mechanisms, or treatment decisions.
  Examples:
  "What is the appropriate treatment for this condition?"
  "How does this imaging pattern affect the patient's prognosis?"

- **2:** High semantic grounding; requires reasoning about specific diseases or well-defined diagnostic entities. Substantial medical knowledge is needed.
  Example:
  "What diseases are included in the image?"

- **3:** Moderate semantic grounding; requires linking features to broad categories of pathology, such as distinguishing between growth, inflammation, or degeneration.
  Example:
  "Do the changes suggest a long-standing damage?"

- **4:** Low–moderate semantic grounding; requires recognition of more specific medical descriptors, but does not involve broad pathology categories or diagnostic reasoning.
  Examples:
  "Does the structure appear to be pushing against or displacing nearby tissues?"
  "Is there a region that appears more diffuse rather than well-demarcated?"

- **5:** Low semantic grounding requires only basic clinical or anatomical recognition (e.g., body parts, organs, simple structures, fractures, nodules).
  Examples:
  "Does the bone show a visible fracture line?"
  "Is there a nodule in this region?"

Therefore, a rating of 3 represents acceptable threshold across all three dimensions: the sample is clinically relevant, visually grounded, and does not require deep semantic knowledge.

During the benchmark curation process, all samples receiving any score below 3 were discarded. Consequently, every VGMED sample satisfies 3 or above on all criteria. This ensured that retained samples genuinely test visual grounding rather than medical reasoning.

Furthermore, as summarized in Tab. N.6, the vast majority of clinician ratings are in the upper categories (4–5), with only a minor proportion of samples receiving a rating of 3 across any axis.

Table N.6: Percentage distribution of clinician ratings (3–5) across all axes for Attribute and Localization questions.

| Type | Category | Rating 3 (%) | Rating 4 (%) | Rating 5 (%) |
|------|----------|--------------|--------------|--------------|
| Attribute | Clinical Relevance | 3.31 | 4.11 | 92.58 |
| | Min. Semantic Grounding | 0.37 | 10.38 | 89.25 |
| | Visual Grounding | 4.04 | 12.18 | 83.77 |
| Localization | Clinical Relevance | 0.02 | 0.52 | 99.46 |
| | Min. Semantic Grounding | 0.05 | 5.76 | 94.19 |
| | Visual Grounding | 3.96 | 11.79 | 84.25 |

## O  LIMITATIONS

While our work provides a systematic and detailed investigation into visual grounding as a key failure mode in medical MLLMs, it focuses exclusively on this aspect. We do not examine other potential sources of failure, such as deficiencies in semantic grounding or reasoning capabilities. In practice, failures may also arise from an inability to recognize what clinical concepts are relevant or to integrate multimodal information effectively. Additionally, our proposed method, VGRefine, is designed to improve visual grounding at inference time but does not address other underlying limitations, such as dataset biases or insufficient domain-specific knowledge. Future work will explore complementary methods to assess and improve semantic grounding and extend our analysis framework to uncover other failure modes.

## P  EXPERIMENTAL SETTING/DETAILS AND COMPUTING RESOURCES

For both VGRefine-7B and VGRefine-34B, we select the top 20 attention heads—ranked by alignment with visually relevant regions—and average their outputs to obtain the filtered attention map. A percentile threshold of 50% is used to suppress low-activation regions during attention knockout. For VGRefine-7B, the attention knockout is applied at layer $l = 16$, while for VGRefine-34B, it is applied at layers $l = 34, 35, 36$, identified through our quantitative analysis as most relevant to visual grounding. We follow a zero-shot evaluation protocol across six biomedical VQA benchmarks: VQA-RAD, SLAKE, PathVQA, PMC-VQA, OmniMedVQA, and MMMU (Health & Medicine track). The full set of prompts used for zero-shot evaluation is provided in Section F.2. All experiments are conducted on a server with 8×NVIDIA A100 80GB GPUs.

## Q  BROADER IMPACTS AND ETHICAL CONSIDERATIONS

This work involves the analysis of medical MLLMs using publicly available datasets that are de-identified. No private or sensitive patient data is used. We acknowledge that the deployment of medical MLLMs carries potential risks, including misinterpretation of clinical images, over-reliance on automated outputs by clinicians, and disparities in performance across patient populations. Our work aims to mitigate such risks by improving the reliability of model predictions through better visual grounding. To promote transparency and reproducibility, we provide open access to code, evaluation metrics, and the VGMED dataset. This enables the broader research community to scrutinize and build upon our work responsibly.

## R  SAFEGUARDS

Our study does not involve training or releasing a new foundation model, but rather evaluates and analyzes existing medical MLLMs in terms of their visual grounding behavior. While our proposed inference-time refinement method improves grounding performance, it is designed for research use only and does not replace expert validation. We do not claim clinical applicability, and no components of our work should be used for medical diagnosis or decision-making without extensive clinical evaluation.

If any models or code are released, access will be gated under a research-use license, and accompanied by usage guidelines clearly stating that they are intended solely for non-commercial, academic use. The evaluation dataset we construct contains only de-identified medical images drawn from publicly available datasets, and all visual content has been reviewed to ensure it does not pose safety, privacy, or dual-use risks.

## S  LICENSES

All datasets and models used in this work are publicly available and cited appropriately in the main paper. We do not scrape any new data from the web or repackage any existing datasets; all visual assets have been used in accordance with their licenses.

## T    USE OF LARGE LANGUAGE MODELS (LLMS)

LLMs were used solely as a writing aid to improve clarity, grammar, and style. They were not involved in generating research ideas, designing methodology, analyzing data, or drawing conclusions.

