# OpenReview forum: "How Do Medical MLLMs Fail?  A Study on Visual Grounding in Medical Images"
_ICLR.cc/2026/Conference — ICLR 2026 Poster_

### Official Review · Reviewer_Bv76 · 2025-10-19

**Soundness:** 2
**Presentation:** 3
**Contribution:** 2
**Rating:** 6
**Confidence:** 3

**Summary:**

This paper investigates the failure modes of medical MLLMs, focusing on their visual grounding ability in Med-VQA. The authors introduce a diagnostic framework that evaluates attention alignment between model-generated attention maps and annotated regions of interest using three metrics—Attention Ratio (AR), Kullback–Leibler (KL) divergence, and Jensen–Shannon (JS) divergence—on the proposed VGMED benchmark. They further present VGRefine, a two-step training-free method that refines attention through (1) filtering low-confidence visual regions to create a high-confidence binary mask and (2) applying this mask to suppress irrelevant visual attention. Experiments on multiple Med-VQA benchmarks show improved grounding alignment and accuracy, with qualitative visualizations and expert evaluations supporting the method’s interpretability and effectiveness.

**Strengths:**

* The paper identifies a clear and meaningful problem—the weak visual grounding ability of medical MLLMs—and addresses it through a structured, quantitative framework. It turns a qualitative interpretability challenge into measurable alignment metrics and integrates them within a unified evaluation setting.

* The proposed inference-time method, VGRefine, is concise, computationally efficient, and easy to deploy. Despite its simplicity, it consistently enhances grounding quality and answer accuracy across diverse Med-VQA benchmarks, demonstrating both technical soundness and practical relevance.

* The experimental section is extensive, including evaluations on multiple datasets, ablation studies, and expert assessments. The combination of quantitative metrics and human evaluation adds credibility and depth to the analysis.

**Weaknesses:**

* The paper attributes model failure primarily to weak visual grounding but does not clearly disentangle it from semantic grounding or reasoning errors. It remains unclear which factor—semantic or visual—dominates the poor generalization of current medical MLLMs. Providing representative failure cases could clarify whether errors stem from misunderstanding the question or mislocalizing visual evidence.

* The choice to compute attention maps only from the last input text token lacks justification. Averaging or weighting attention across all input tokens might yield a more stable and comprehensive alignment estimation; offering an explanation or comparison for this design choice would strengthen the analysis.

* Attention Knockout is applied at a single layer, yet the broader influence on higher layers or cross-layer propagation of redundant visual information is not analyzed, which may limit the consistency of the refinement effect.

* Comparisons with general-domain vision-language models are limited. Including stronger or more recent baselines, such as Qwen2.5-VL or InternVL3, could better support the argument that grounding difficulty is domain-inherent rather than model-specific.

* While the study provides a thorough diagnostic view, it remains primarily descriptive. Further exploration into why grounding failures arise—such as biases in medical datasets or insufficient visual diversity—would deepen the explanatory value of the work.

**Questions:**

* Provide an explanation for using only the last input text token to compute attention maps. It would be helpful to test or discuss whether averaging or weighting attention across all input tokens offers a more reliable alignment estimation.

* Explain the rationale for applying Attention Knockout at a single layer (e.g., layer 16) and whether extending it to multiple or higher layers could further improve grounding consistency.

* Consider including newer or stronger vision-language models (such as Qwen2.5-VL or InternVL) in the comparison to confirm whether grounding difficulty is indeed domain-specific.

* Discuss possible training-related causes of grounding failure, such as dataset bias or limited visual diversity, and whether VGRefine could mitigate these underlying issues.

---

> ### Author Response · Authors · 2025-11-21
> **[Response to Reviewer Bv76] Part 1/5**
>
> We appreciate the positive feedback and valuable comments of the reviewer. We sincerely hope that the reviewer could consider increasing the ratings if our responses have addressed all questions.
>
> $ $
>
> >**W1**:  The paper attributes model failure primarily to weak visual grounding but does not clearly disentangle it from semantic grounding or reasoning errors. It remains unclear which factor—semantic or visual—dominates the poor generalization of current medical MLLMs. Providing representative failure cases could clarify whether errors stem from misunderstanding the question or mislocalizing visual evidence.
>
> **Our Answer**:
>
> We thank the reviewer for this insightful question. We clarify that our paper does not attribute medical MLLM failure primarily to weak visual grounding. Rather, we position inadequate grounding as one important and previously underexplored failure mode, alongside other issues such as semantic grounding and reasoning errors. This is made explicit in the Abstract, where we state:
>
> > “Overall, our work, for the first time, systematically validates inadequate visual grounding as one of the key contributing factors for medical MLLMs' under-performance.”
>
> Furthermore, VGMED is the first diagnostic dataset specifically designed to disentangle visual grounding from semantic grounding in medical MLLMs. Existing medical VQA datasets and prior studies do not separate these factors, making it difficult to determine whether failures arise from semantic misunderstanding or mislocalized visual evidence. In contrast, VGMED was carefully co-created with three clinicians to ensure that all samples require reference to localized, clinically annotated regions without requiring deep semantic or medical knowledge (Sec. 2.1). We also provide detailed annotation statistics in Sec. R in the manuscript (Sec. B in Supp for Rebuttal). This enables the first systematic study of visual grounding behavior in medical MLLMs.
>
> Thus, our analysis isolates visual grounding from semantic errors by construction. Following reviewer's suggestion, we have added representative failure cases in Fig. Q27 (Fig. 4 in Supp for Rebuttal) illustrating: (i) semantic misunderstanding and (ii) visual mislocalization. These examples clarify that while multiple factors contribute to poor generalization, weak visual grounding consistently emerges as a major and measurable issue, though not the sole cause.
>
>
> $ $
>
>
> (continue in Part 2)

---

> > ### Author Response · Authors · 2025-11-21
> > **[Response to Reviewer Bv76] Part 2/5**
> >
> > >**W2**:  The choice to compute attention maps only from the last input text token lacks justification. Averaging or weighting attention across all input tokens might yield a more stable and comprehensive alignment estimation; offering an explanation or comparison for this design choice would strengthen the analysis.
> >
> > >**Q1**:  Provide an explanation for using only the last input text token to compute attention maps. It would be helpful to test or discuss whether averaging or weighting attention across all input tokens offers a more reliable alignment estimation.
> >
> >
> > **Our Answer**:
> >
> > Thank you for your comment. We clarify that our choice to compute attention maps from the last input text token follows the standard practice in prior visual grounding work on general-domain MLLMs [1,2].
> >
> > In decoder-style LLMs, each generated (or processed) token attends to all previous tokens; consequently, the query vector of the last input token encodes the contextualized semantics of the entire sentence [1,2]. By the time the model reaches the final token of a prompt or question, it has already aggregated information from all preceding words. Kang et al. [1], for example, explicitly argue that this final token can serve as a “representative query” for the whole sentence, encapsulating its full context. We therefore adopt the last input token attention as the grounding signal to remain consistent with prior general-domain studies.
> >
> > Importantly, adopting the same approach as in prior visual grounding work on general-domain studies enables **direct comparison** and contrast with our key finding that such grounding often fails in medical images.
> >
> >
> >
> > To further validate the effectiveness of this design choice, we additionally conduct an ablation study evaluating alignment quality when computing attention maps from:
> >
> > * the average over all text tokens,
> > * the average over question tokens only
> >
> >
> > while keeping all other settings identical to Sec. 2. We evaluate both a representative general domain MLLM LLaVA on COCO images, and a SOTA medical MLLM HuatuoGPT-Vision on both VGMED and COCO. Please see the results in Fig. Q28 (Fig. 5 in Supp for Rebuttal)
> >
> > From this ablation study, we confirm that attention maps computed from the last input token achieves equal or better alignment with ground-truth regions compared to the alternative options. This observation aligns with the design choice made by previous work.
> >
> > We will include this discussion in the final version to provide a more complete justification of our design.
> >
> >
> > $ $
> >
> >
> > [1]. Kang, Seil, et al. "Your large vision-language model only needs a few attention heads for visual grounding." Proceedings of the Computer Vision and Pattern Recognition Conference. 2025.
> >
> > [2]. Zhang, Jiarui, et al. "MLLMs Know Where to Look: Training-free Perception of Small Visual Details with Multimodal LLMs." The Thirteenth International Conference on Learning Representations. 2025.
> >
> >
> > $ $
> >
> >
> > (continue in Part 3)

---

> ### Author Response · Authors · 2025-11-21
> **[Response to Reviewer Bv76] Part 3/5**
>
> >**W3**:  Attention Knockout is applied at a single layer, yet the broader influence on higher layers or cross-layer propagation of redundant visual information is not analyzed, which may limit the consistency of the refinement effect.
>
> >**Q2**:  Explain the rationale for applying Attention Knockout at a single layer (e.g., layer 16) and whether extending it to multiple or higher layers could further improve grounding consistency.
>
>
> **Our Answer**:
>
> Thank you for your comment. We performed an ablation study on modifying more layers with attention knockout. More specifically, we varied the window size of modified layers centered around the layer most relevant to visual grounding, as identified by our KL divergence analysis. Our result shows that modifying more layers doesn’t significantly affect the performance. Therefore, for simplicity, we apply VGRefine at the layer with the highest relevance to visual grounding.
>
> #### Table C1. Ablation study on number of layers to be modified with attention knockout during VGRefine.
> | NO. of Layers  | VQA-RAD | SLAKE | PathVQA | PMC-VQA | Avg. |
> |---|---|---|---|---|---|
> | 1  | **71.24** | 76.88 | **67.61** | **56.20** | **68.42** |
> | 3  | 70.62 | 77.02 | 67.25 | 55.95 | 68.20 |
> | 5  | 70.86 | 77.27 | 67.13 | 55.55 | 68.18 |
> | 7  | 71.09 | 77.74 | 66.81 | 55.45 | 68.21 |
> | 9 | 70.93 | 77.96 | 66.81 | 55.65 | 68.29 |
> | 11 | 71.22 | **78.14** | 66.42 | 55.55 | 68.24 |
>
>
> $ $
>
>
> (continue in Part 4)

---

> > ### Author Response · Authors · 2025-11-21
> > **[Response to Reviewer Bv76] Part 4/5**
> >
> > >**W4**: Comparisons with general-domain vision-language models are limited. Including stronger or more recent baselines, such as Qwen2.5-VL or InternVL3, could better support the argument that grounding difficulty is domain-inherent rather than model-specific.
> >
> > >**Q3**:  Consider including newer or stronger vision-language models (such as Qwen2.5-VL or InternVL) in the comparison to confirm whether grounding difficulty is indeed domain-specific
> >
> > **Our Answer**:
> >
> >
> > Thank you for the question. Following reviewer’s suggestion, we include the latest general MLLMs: Qwen2.5-VL-7B and InternVL3-8B. Please see the results in Fig. Q26 (Fig. 3 in Supp for Rebuttal). We observe that the visual grounding deficiency persists even in these latest general-purpose models.
> >
> >
> > We further emphasize that our study explicitly investigates whether grounding failures arise because medical MLLMs are inherently weaker models or because the issue is specific to medical images. As shown in Figure 3 (main paper), the same medical MLLMs (e.g. HuatuoGPT-V 7B, MedRegA 34B) that exhibit poor grounding on medical images demonstrate clear and coherent grounding when applied to natural RGB images (MS-COCO). This cross-domain evaluation indicates that **the grounding deficiency is not due to model weakness, but rather that medical images introduce domain-inherent grounding challenges that current medical MLLMs do not yet adequately address.**
> >
> >
> > $ $
> >
> >
> > (continue in Part 5)

---

> > > ### Author Response · Authors · 2025-11-21
> > > **[Response to Reviewer Bv76] Part 5/5**
> > >
> > > >**W5**: While the study provides a thorough diagnostic view, it remains primarily descriptive. Further exploration into why grounding failures arise—such as biases in medical datasets or insufficient visual diversity—would deepen the explanatory value of the work.
> > >
> > > >**Q4**:  Discuss possible training-related causes of grounding failure, such as dataset bias or limited visual diversity, and whether VGRefine could mitigate these underlying issues.
> > >
> > > **Our Answer**:
> > >
> > > Thank you for the thoughtful comment. We agree with the reviewer that dataset bias and limited visual diversity are important potential contributors to grounding failures in medical MLLMs. More broadly, we believe that inadequate visual grounding in medical MLLMs may also arise from several training-related factors, including the aforementioned dataset bias and visual diversity issues, as well as the limited availability of high-quality paired data, insufficient cross-modal alignment during training, and the substantial heterogeneity across medical imaging modalities.
> > >
> > >
> > > Here, we highlight a key potential reason for inadequate grounding on medical images: **excessive intra-medical variability** in how medical visual data are internally represented. Although current MLLMs treat all medical imaging types (CT, MRI, X-ray, ultrasound, etc.) as a single unified modality during training, our analysis shows that the model in fact encodes these modalities as distinct and heterogeneous distributions.
> > >
> > > If medical modalities formed a coherent domain, distances among CT/MRI/ultrasound/etc. representations should be substantially smaller than the distances between medical and natural images. However, quantitative evidence using the Modality Integration Rate (MIR) metric [1] contradict this assumption. Specifically, in HuatuoGPT-Vision:
> > >
> > > * the average MIR across medical modalities is 20.19
> > > * the MIR between medical and general-domain data is 23.49
> > >
> > > See results in Tab. C2. This indicates that the model treats CT, MRI, X-ray, ultrasound, etc. as nearly as different from each other as they are from natural images, despite all being trained as if they belonged to one modality.
> > >
> > > **Such excessive intra-medical variability makes it difficult for the model to learn a stable and unified mapping between textual concepts and visual features.** For example, the liver appears fundamentally different in CT (attenuation patterns), MRI (T1/T2 signal characteristics), and ultrasound (echogenicity and speckle texture), leading the model to encode "liver" as multiple far-apart visual clusters rather than a coherent manifold. As a result, the language token "liver" cannot be linked to a single, stable visual prototype; instead, the model must implicitly map one textual concept to several incompatible visual clusters. This fragmented alignment makes grounding noisy and unstable, leading to diffuse or incorrect localization even for basic anatomical structures such as the liver.
> > >
> > > We will integrate this analysis into the final version to provide a more complete discussion of training-related causes underlying grounding failures.
> > >
> > >
> > > #### Table C2. MIR between medical domains in the embedding space of SoTA medical MLLM HuatuoGPT-Vision
> > > |        | MRI   | CT    | MI    | DP    | US    | Others | IRI  |
> > > |--------|-------|-------|-------|-------|-------|--------|--------|
> > > | MRI    |       |       |       |       |       |        |        |
> > > | CT     | 12.30 |       |       |       |       |        |        |
> > > | MI     | 19.69 | 18.76 |       |       |       |        |        |
> > > | DP     | 26.29 | 25.59 | 20.45 |       |       |        |        |
> > > | US     | 22.22 | 21.87 | 22.27 | 25.17 |       |        |        |
> > > | Others | 18.53 | 16.14 | 18.36 | 17.45 | 20.00 |        |        |
> > > | IRI    | 18.89 | 17.52 | 16.03 | 21.13 | 20.76 | 17.47  |        |
> > > | Endo   | 23.38 | 19.40 | 19.86 | 18.62 | 26.84 | 22.02  | 18.36  |
> > >
> > >
> > > $ $
> > >
> > >
> > > VGRefine, as a training-free method, does not directly resolve the underlying training-related issues that emerge during model pretraining. However, **its strength lies in mitigating the effects of these issues at inference time**. As demonstrated in Sec. 4.2 and Supp. E, G, VGRefine improves visual grounding by identifying the model’s component that best aligns with visual grounding on clinically relevant regions and suppressing spurious or irrelevant attention. Our human evaluation among clinicians in Sec. 4.3 further demonstrates that our proposed VGRefine enhances clinician trust by producing more interpretable visual features.
> > >
> > >
> > > $ $
> > >
> > >
> > > [1] Huang Q, et al. "Deciphering Cross-Modal Alignment in Large Vision-Language Models via Modality Integration Rate." Proceedings of the IEEE/CVF International Conference on Computer Vision. 2025.

---

> ### Comment · Reviewer_Bv76 · 2025-11-27
>
> I appreciate the detailed explanations and additional experiments provided in the authors’ response. Overall, the authors have addressed my questions regarding the method’s motivation, interpretability, design choices, and potential training factors with sufficient and well-reasoned clarifications. The newly added ablation studies and more comprehensive visualization examples also strengthen the analysis. Moreover, the expanded cross-model comparisons and cross-domain analyses further enhance the credibility of the arguments.
>
> In general, while the paper makes meaningful contributions in proposing a diagnostic framework and offering improved interpretability, the current method still leans more toward diagnosis and inference rather than providing a solution that fundamentally enhances model capability. Therefore, after considering all aspects, I have decided to maintain my original positive rating.

---

> > ### Author Response · Authors · 2025-11-28
> > **Thank you Reviewer Bv76 for the thoughtful follow-up and the positive rating**
> >
> > We sincerely thank the reviewer for the thoughtful follow-up and the positive rating. We appreciate the recognition of our additional analyses and strengthened cross-model and cross-domain experiments, which further reinforce the soundness of our methodology and the credibility of our findings.
> >
> >
> > We respectfully hope the reviewer may consider reflecting these improvements in the **soundness and contribution ratings**. Thank you again for your constructive feedback and support.

---

### Official Review · Reviewer_2uoh · 2025-10-30

**Soundness:** 3
**Presentation:** 3
**Contribution:** 3
**Rating:** 6
**Confidence:** 4

**Summary:**

This work presents VGMED, a clinician-curated benchmark for evaluating visual grounding in medical MLLMs, and proposes VGRefine, an inference-time attention refinement strategy that significantly improves visual grounding and zero-shot Med-VQA performance. The authors systematically analyze failure modes in existing medical MLLMs and demonstrate that inadequate visual grounding is a major bottleneck. The study is well-motivated, carefully designed, and supported by strong empirical results across multiple tasks and modalities.

**Strengths:**

1. The paper isolates visual grounding as a distinct bottleneck in medical multimodal models, a perspective that is under-explored yet highly relevant for trustworthy deployment.
2. Layer- and head-specific analysis of cross-attention provides an interpretable pathway to understanding failure behavior, in line with mechanistic interpretability trends in multimodal research.
3. The inference-time attention refinement is conceptually clean, model-agnostic, and introduces no additional data or parameters. The performance gains across diverse modalities and benchmarks are substantial.

**Weaknesses:**

1. The reliance on attention as the primary grounding signal may be discussed more rigorously; a short reflection on alternative grounding indicators (e.g., gradient-based saliency, causal perturbation) would improve completeness.

2. The benchmark curation process is strong, but clearer quantitative annotation statistics (e.g. agreement checks, diversity distribution) would further support dataset rigor.

3. Method hyperparameters (selected heads/layers) are fixed; small discussion of adaptive or data-driven selection would enhance generality claims.

**Questions:**

See Weaknesses.

---

> ### Author Response · Authors · 2025-11-21
> **[Response to Reviewer 2uoh] Part 1/3**
>
> We appreciate the positive feedback and valuable comments of the reviewer. We sincerely hope that the reviewer could consider increasing the ratings if our responses have addressed all questions.
>
> $ $
>
> >**Q1**:  The reliance on attention as the primary grounding signal may be discussed more rigorously; a short reflection on alternative grounding indicators (e.g., gradient-based saliency, causal perturbation) would improve completeness.
>
> **Our Answer**:
>
> Thank you for your comment. We fully agree that discussing alternative grounding indicators can improve completeness. We clarify that we adopt attention maps as the primary grounding signal for the following reasons:
> * Attention maps have been frequently adopted in recent work on visual grounding in general-domain MLLMs [1,2,3]. Importantly, a closely related study [1] used attention to show that MLLMs can ground effectively in natural images. By adopting a similar approach, our study enables direct comparison and contrast—highlighting our key finding that visual grounding often fails in the medical images.
> * We were also able to reproduce and validate the results from this prior work, confirming that attention maps clearly reflect proper visual grounding in natural scenes. This contrast makes the deficiencies in medical MLLMs more striking and informative.
>
> Following reviewer’s suggestion, in the updated manuscript, we will add a short reflection on alternative grounding indicators:
>
> * Gradient-based saliency, which requires backpropagation, is more computation-intensive compared to attention maps.
> * Causal perturbation techniques demand a new forward pass for each perturbed input (e.g., region masking or token removal), making them computationally prohibitive for large-scale medical grounding analysis.
>
>
>
> $ $
>
>
>
> [1]. Zhang, Jiarui, et al. "MLLMs Know Where to Look: Training-free Perception of Small Visual Details with Multimodal LLMs." The Thirteenth International Conference on Learning Representations. 2025.
>
> [2]. Kang, Seil, et al. "Your large vision-language model only needs a few attention heads for visual grounding." Proceedings of the Computer Vision and Pattern Recognition Conference. 2025.
>
> [3]. Kaduri O, et al. "What's in the Image? A Deep-Dive into the Vision of Vision Language Models.” Proceedings of the Computer Vision and Pattern Recognition Conference. 2025.
>
>
> $ $
>
>
> (continue in Part 2)

---

> ### Author Response · Authors · 2025-11-21
> **[Response to Reviewer 2uoh] Part 2/3**
>
> >**Q2**:  The benchmark curation process is strong, but clearer quantitative annotation statistics (e.g. agreement checks, diversity distribution) would further support dataset rigor.
>
>
> **Our Answer**:
>
>
> We thank the reviewer for the insightful comment and welcome the opportunity to further highlight the rigor of our benchmark curation process. Ensuring annotation quality was a central focus in constructing VGMED.
>
> We now provide quantitative annotation statistics during the benchmark curation process.
>
> As described in the main paper, three certified medical doctors contributed to VGMED by (Sec. 2.1):
>
> (1) co-designing GPT prompts to elicit clinically meaningful and visually grounded questions,
>
> (2) reviewing and refining all samples for clinical relevance and grounding focus, and
>
> (3) verifying that every sample requires reference to the annotated region without requiring deep semantic or diagnostic knowledge.
>
> During curation, clinicians graded each sample on three standardized 1–5 scales: Clinical Relevance, Visual Grounding, and Minimum Semantic Grounding. A brief summary is provided below, with full definitions included in Sec. R in the manuscript (Sec. B in Supp for Rebuttal).
>
>
> $ $
>
>
> Clinical Relevance (1–5):
>
> • 1–2: irrelevant or marginal medical value
>
> • 3 = “Acceptable” — the question is reasonable in clinical significance
>
> • 4–5: clearly useful or highly valid clinical questions
>
>
> $ $
>
>
> Visual Grounding (1–5):
>
> • 1–2: weak or no correspondence to the annotated region
>
> • 3 = “Reasonably aligned” — the question overlaps with or implies the boxed region
>
> • 4–5: clear or perfect correspondence
>
>
> $ $
>
>
> Minimum Semantic Grounding (1–5):
>
> • 1–2: requires substantial medical knowledge or diagnostic reasoning
>
> • 3 = “Moderate semantic grounding” — requires only broad pathology distinctions, not deep disease-specific expertise
>
> • 4–5: minimal semantic grounding
>
>
> $ $
>
>
> A rating of 3 thus represents acceptable threshold across all three dimensions: the sample is clinically relevant, visually grounded, and does not require deep semantic knowledge.
>
> During the benchmark curation process, all samples receiving any score below 3 were discarded. Consequently, every VGMED sample satisfies ≥3 on all criteria. This filtering ensured that all VGMED samples genuinely test visual grounding rather than medical reasoning.
>
>
> Furthermore, the majority achieves the highest score (5):
>
>
> $ $
>
>
> Clinical Relevance: 96.02%
>
> Visual Grounding: 84.01%
>
> Minimum Semantic Grounding: 91.72%
>
>
> $ $
>
>
> We also provide average ratings for localization and attribute questions in VGMED:
> #### Table B1. Average ratings for localization and attribute questions
>
> | Rating Aspect                  | Attribute Q. | Localization Q. |
> |---------------------------|--------------|------------------|
> | Clinical Relevance        | 4.89         | 4.99             |
> | Visual Grounding          | 4.8         | 4.8             |
> | Minimum Semantic Grounding| 4.89         | 4.94             |
>
> Additional annotation statistics and more details are included in the Sec. R in the manuscript (Sec. B in Supp for Rebuttal). These results demonstrate the care taken to ensure VGMED is a rigorous and clinically validated benchmark for analyzing visual grounding in medical MLLMs.
>
>
> $ $
>
>
> (continue in Part 3)

---

> ### Author Response · Authors · 2025-11-21
> **[Response to Reviewer 2uoh] Part 3/3**
>
> >**Q3**:  Method hyperparameters (selected heads/layers) are fixed; small discussion of adaptive or data-driven selection would enhance generality claims.
>
> **Our Answer**:
>
> Thank you for the thoughtful comment. We clarify that the selection of heads and layers in our method is **data-driven** rather than manually fixed, as described in Sec. 4.1 and Supp. D.
>
> Specifically, we quantify the visual-grounding relevance of every attention head and layer by measuring the alignment between their attention distributions and ground-truth annotations in our 28K-sample VGMED dataset, which covers eight imaging modalities and six anatomical regions.
>
> We further highlight that, to avoid any risk of data leakage, our head- and layer-selection procedure is performed using randomly sampled MS-COCO images, as illustrated in Fig. D.1. This choice is justified by our empirical finding that the attention heads most relevant to visual grounding in natural scene images are often also the most relevant for medical images. This cross-domain consistency enables us to identify visually robust heads and layers without touching any medical samples.
>
> By selecting the components that demonstrate the highest grounding alignment (lowest KL divergence) in this data-driven manner, our method naturally adapts to the intrinsic attention structure of the model. Given that the selection is derived from a diverse, medical modality-agnostic corpus (COCO), and that these heads consistently generalize well to medical images, we believe this approach supports the generality claims of VGRefine, in addition to our experiment results across 6 diverse Med-VQA benchmarks (over 110K VQA samples from 8 imaging modalities) in Sec. 4.2.

---

> > ### Author Response · Authors · 2025-11-28
> > **Thank you again for your thoughtful feedback**
> >
> > Dear Reviewer 2uoh:
> >
> > Thank you again for your thoughtful feedback. As the discussion period is ending soon, we would greatly appreciate hearing whether our rebuttal adequately addressed your questions.
> >
> > We are happy to respond promptly to any additional questions you may have.

---

### Official Review · Reviewer_Qozi · 2025-11-01

**Soundness:** 3
**Presentation:** 3
**Contribution:** 3
**Rating:** 4
**Confidence:** 4

**Summary:**

The paper investigates why medical multimodal large language models (MLLMs) underperform in medical image interpretation tasks, particularly in zero-shot settings where generalization is critical.
Through an empirical analysis of eight state-of-the-art medical MLLMs, the authors demonstrate that these models often fail to ground their predictions in clinically relevant image regions—a deficiency that appears specific to medical imaging. In contrast, prior studies show that general-domain MLLMs exhibit appropriate visual grounding when applied to natural image understanding tasks.
Motivated by this finding, the paper introduces VGRefine, a simple yet effective inference-time method that optimizes the model’s attention distribution to improve visual grounding and reasoning in medical contexts. The proposed method achieves state-of-the-art results on six Med-VQA benchmarks covering eight imaging modalities and more than 110,000 samples.

**Strengths:**

1. The study is interesting. By examining whether MLLMs can reuse their reasoning attention for localization, the authors observe that models’ attention during question answering is not fully concentrated on relevant image regions, and finally propose VGRefine to improve model performance.

2. The paper is the first to systematically verify that insufficient visual grounding is one of the key factors leading to poor performance of medical MLLMs. The proposed method achieves state-of-the-art results on six different Med-VQA benchmarks covering eight imaging modalities and more than 110,000 VQA samples.

**Weaknesses:**

1. The paper could include more mathematical analysis to explain the causes behind the observed phenomenon.

2. The paper claims that this phenomenon has not been observed in other domains. Do “other domains” refer only to general visual question answering on natural scenes? Or does the paper merely show results on some general scenes? If so, can this truly prove that the same phenomenon does not appear in any other specialized domains?

3. The models used in the experiments—such as Qwen-VL, LLaVA-Med, and LLaVA-1.5—are relatively outdated. On the latest visual models, whether domain-specific or general-purpose, can the observed phenomenon and the effectiveness of the proposed method still hold? Could the authors also provide experiments showing how medical models perform in general VQA scenarios?

**Questions:**

1. What do the authors believe are the reasons why this phenomenon occurs in medical scenarios?

2. In Figure 1 of the main text, the paper compares several domain-specific medical models on medical VQA tasks with LLaVA-1.5 on general VQA tasks. What if this comparison were reversed? Is the problem caused by these medical models themselves? If the proposed method were directly applied to stronger and more recent general MLLMs, such as Qwen2.5-VL or Qwen3-VL, would the same conclusions be obtained?

If the authors can address the above questions, the reviewer promises to raise the score.

---

> ### Author Response · Authors · 2025-11-21
> **[Response to Reviewer Qozi] Part 1/4**
>
> We appreciate the positive feedback and valuable comments of the reviewer. We sincerely hope that the reviewer could consider increasing the ratings if our responses have addressed all questions.
>
> $ $
>
> >**W1**: The paper could include more mathematical analysis to explain the causes behind the observed phenomenon.
>
> >**Q1**:  What do the authors believe are the reasons why this phenomenon occurs in medical scenarios?
>
> **Our Answer**:
>
> Thank you for the thoughtful comment. We believe that inadequate visual grounding in medical MLLMs may stem from several factors, including limited availability of high-quality paired data, insufficient cross-modal alignment during training, and significant heterogeneity across medical imaging modalities.
>
> Here, we highlight a key potential reason for inadequate grounding on medical images: **excessive intra-medical variability** in how medical visual data are internally represented. Although current MLLMs treat all medical imaging types (CT, MRI, X-ray, ultrasound, etc.) as a single unified modality during training, our analysis shows that the model in fact encodes these modalities as distinct and heterogeneous distributions.
>
> If medical modalities formed a coherent domain, distances among CT/MRI/ultrasound/etc. representations should be substantially smaller than the distances between medical and natural images. However, quantitative evidence using the Modality Integration Rate (MIR) metric [1] contradict this assumption. Specifically, in HuatuoGPT-Vision:
>
> * the average MIR across medical modalities is 20.19
> * the MIR between medical and general-domain data is 23.49
>
> See results in Tab. A1. This indicates that the model treats CT, MRI, X-ray, ultrasound, etc. as nearly as different from each other as they are from natural images, despite all being trained as if they belonged to one modality.
>
> **Such excessive intra-medical variability makes it difficult for the model to learn a stable and unified mapping between textual concepts and visual features.** For example, the liver appears fundamentally different in CT (attenuation patterns), MRI (T1/T2 signal characteristics), and ultrasound (echogenicity and speckle texture), leading the model to encode "liver" as multiple far-apart visual clusters rather than a coherent manifold. As a result, the language token "liver" cannot be linked to a single, stable visual prototype; instead, the model must implicitly map one textual concept to several incompatible visual clusters. This fragmented alignment makes grounding noisy and unstable, leading to diffuse or incorrect localization even for basic anatomical structures such as the liver.
>
> We will integrate this analysis into the final version to provide a more complete discussion of training-related causes underlying grounding failures.
>
> $ $
>
> #### Table A1. MIR between medical domains in the embedding space of HuatuoGPT-Vision
> |        | MRI   | CT    | MI    | DP    | US    | Others | IRI  |
> |--------|-------|-------|-------|-------|-------|--------|--------|
> | MRI    |       |       |       |       |       |        |        |
> | CT     | 12.30 |       |       |       |       |        |        |
> | MI     | 19.69 | 18.76 |       |       |       |        |        |
> | DP     | 26.29 | 25.59 | 20.45 |       |       |        |        |
> | US     | 22.22 | 21.87 | 22.27 | 25.17 |       |        |        |
> | Others | 18.53 | 16.14 | 18.36 | 17.45 | 20.00 |        |        |
> | IRI    | 18.89 | 17.52 | 16.03 | 21.13 | 20.76 | 17.47  |        |
> | Endo   | 23.38 | 19.40 | 19.86 | 18.62 | 26.84 | 22.02  | 18.36  |
>
> $ $
>
> [1] Huang Q, Dong X, Zhang P, et al. "Deciphering Cross-Modal Alignment in Large Vision-Language Models via Modality Integration Rate.” Proceedings of the IEEE/CVF International Conference on Computer Vision. 2025.
>
> $ $
>
> (continue in Part 2)

---

> ### Author Response · Authors · 2025-11-21
> **[Response to Reviewer Qozi] Part 2/4**
>
> >**Q2**: In Figure 1 of the main text, the paper compares several domain-specific medical models on medical VQA tasks with LLaVA-1.5 on general VQA tasks. What if this comparison were reversed? Is the problem caused by these medical models themselves?
>
> >**W3**: …Could the authors also provide experiments showing how medical models perform in general VQA scenarios?
>
> **Our Answer**:
>
> We thank the reviewer for this thoughtful suggestion. Following the reviewer’s comment, we conducted the reversed comparison:
>
> 1. Domain-specific medical models on general rgb images and general VQA tasks
> 2. General domain model LLaVA-v1.5 on medical images and medical VQA tasks  (more results of SOTA general domain models in the next part)
>
> Please see Fig. Q24 in manuscript (Fig.1 in Supp for Rebuttal) for qualitative examples, including HuatuoGPT-V, LLaVA-Tri and MedRegA on general domain images (from COCO), and LLaVA-v1.5 on medical images (from VGMED). In Fig. Q25 (Fig.2 in Supp for Rebuttal), we additionally include quantitative evaluation of LLaVA-v1.5 on medical images.
>
> **The results fully align with and further strengthen our core claim.** Specifically, we observe that:
>
> 1. General-domain models fail to ground their predictions in clinically relevant regions when operating on medical images and medical VQA tasks.
>
> 2. Medical-domain models can ground their predictions when applied to natural images and general VQA tasks. In addition to the qualitative results, we kindly note that we already have quantitative grounding analysis in Fig. 3 of the main paper to apply medical models on COCO, which are consistent with these qualitative results. Particularly, as shown in Fig. 3 of our main paper, the SOTA medical MLLM HuatuoGPT-Vision-7B achieves **comparable visual grounding performance on natural scene data** relative to a well-trained general-domain model such as LLaVA-v1.5 (e.g., KL↓: 2.0 vs. 2.2; JS↓: 0.40 vs. 0.40). This, together with the qualitative results, demonstrate that medical MLLMs possess good visual grounding capabilities in general-domain settings. However, when the same medical MLLM is evaluated on medical images, its grounding quality drops substantially (KL↓: 3.1; JS↓: 0.57), **despite being trained on large-scale medical image–text data**.
>
>
> **Overall, this confirms that the grounding failure is not due to model weakness, but is fundamentally specific to the medical domain, consistent with our central findings. Inadequate visual grounding is a medical-domain failure mode.**
>
> We include these new results in the updated manuscript, as they further reinforce our conclusion that inadequate visual grounding is a key factor underlying medical MLLMs’ under-performance.
>
>
> $ $
>
>
> (continue in Part 3)

---

> ### Author Response · Authors · 2025-11-21
> **[Response to Reviewer Qozi] Part 3/4**
>
> >**Q2**: …If the proposed method were directly applied to stronger and more recent general MLLMs, such as Qwen2.5-VL or Qwen3-VL, would the same conclusions be obtained?
>
>
> >**W3**:  The models used in the experiments—such as Qwen-VL, LLaVA-Med, and LLaVA-1.5—are relatively outdated. On the latest visual models, whether domain-specific or general-purpose, can the observed phenomenon and the effectiveness of the proposed method still hold?
>
>
> **Our Answer**:
>
> Thank you for the question. We would like to clarify that **for domain-specific medical MLLMs, we have already included the latest models** in our comprehensive analysis and experiments in the main paper. The results support that the observed phenomenon (i.e., inadequate visual grounding in the medical domain) and the effectiveness of the proposed method (VGRefine) still hold in the latest medical MLLMs.
>
> Particularly, most of the medical MLLMs in our analysis in Sec 2.4 and our experiments in Sec 4 (and Supp) were proposed in less than a year: LLaVA-Tri (Xie et al., 2025), HuatuoGPT-Vision-7B/34B (Chen et al., 2024a) (abbreviated as HuatuoGPT-V), VILA-M3-8B/13B (Nath et al., 2024), MedRegA (Wang et al., 2025).
>
> For general-purpose MLLMs, we follow reviewer’s suggestion to include the latest general MLLMs: Qwen2.5-VL-7B and InternVL3-8B. The observed phenomenon and the effectiveness of the proposed method still hold in these latest general MLLMs:
>
> 1. The visual grounding deficiency persists even in the latest general-purpose models.
> 2. The effectiveness of the proposed method (VGRefine) still holds in these latest general models, and the same conclusions can be obtained when the proposed method is applied to Qwen2.5-VL-7B and InternVL3-8B.
>
>
> $ $
>
>
> **1\) Observed phenomenon still holds in the latest general MLLMs.**
>
> For qualitative evaluation, we present 1) Qwen2.5-VL-7B on VGMED 2) InternVL3-8B on VGMED
>
> For quantitative evaluation, we present 1) Qwen2.5-VL-7B on VGMED and COCO 2) InternVL3-8B on VGMED and COCO
>
> Please see all these evaluation results in Fig. Q26 (Fig. 3 in Supp for Rebuttal).
>
> From our evaluation, we observe that **the visual grounding deficiency persists even in modern general-purpose models.**
>
>
>
> $ $
>
>
>
> **2\) The effectiveness of the proposed method still holds in the latest general MLLMs.**
>
> We further evaluate the effectiveness of our proposed VGRefine on the latest general-domain MLLMs, Qwen2.5-VL and InternVL3, on medical VQA datasets, following the same implementation setup as in the main paper Sec. 4. As shown in Tab. A2, VGRefine consistently improves performance across all base models, underscoring its ability to enhance visual grounding in medical images.
>
> #### Table A2. Effectiveness of the proposed method on latest general domain models
> | Model              | VQA-RAD | SLAKE | PathVQA | PMC-VQA | Avg. |
> |--------------------|---------|-------|---------|---------|------|
> | InternVL3-8B | 75.4 |  59.7 | 60.2 | 50.2 | 60.2 |
> | InternVL3-8B + VGRefine (Ours) | 76.7 |  61.20 | 60.7 | 51.3 | 62.1 |
> | Qwen2.5-VL-7B | 77.4 | 73.5 | 67.8 | 53.4 |  67.7 |
> | Qwen2.5-VL-7B + VGRefine (Ours) | 77.7 | 76.3 | 70.1 | 54.0 |  69.2 |
>
>
>
> $ $
>
>
> (continue in Part 4)

---

> > ### Author Response · Authors · 2025-11-21
> > **[Response to Reviewer Qozi] Part 4/4**
> >
> > >**Q4**: The paper claims that this phenomenon has not been observed in other domains. Do “other domains” refer only to general visual question answering on natural scenes? Or does the paper merely show results on some general scenes? If so, can this truly prove that the same phenomenon does not appear in any other specialized domains?
> >
> > **Our Answer**:
> >
> > Thank you for raising this clarification. In the paper, our intention was not to claim that grounding failures categorically never occur in any other specialized domain. Rather, as stated in the manuscript:
> >
> > > “We note that this finding is specific to medical image analysis; in contrast, prior work has shown that MLLMs are capable of grounding their predictions in the correct image regions when applied to natural images (Zhang et al., 2025a).”
> >
> > Here, “other domains” specifically refers to general natural-image visual grounding benchmarks, such as those evaluated in MS-COCO or other natural-scene VQA datasets in prior work (e.g., Zhang et al., 2025a). Our experiments verify that, in these natural-image settings, models demonstrate substantially stronger alignment between attention maps and annotated regions, in contrast to their behavior on medical images.
> >
> > We agree that demonstrating good grounding performance on natural images does not logically exclude the possibility that similar grounding failures might exist in other highly specialized visual domains (e.g., remote sensing, scientific figures). Therefore, in the revised version, we will clarify that our claim is limited to the contrast between medical images and general natural images, and we do not assert conclusions about all other specialized domains. Instead, our findings may motivate future work to investigate whether similar failures arise in domains that share structural properties with medical imagery.

---

> > > ### Author Response · Authors · 2025-11-28
> > > **Thank you again for your thoughtful feedback**
> > >
> > > Dear Reviewer Qozi:
> > >
> > > Thank you again for your thoughtful feedback. As the discussion period is ending soon, we would greatly appreciate hearing whether our rebuttal adequately addressed your questions.
> > >
> > >
> > > We are happy to respond promptly to any additional questions you may have.

---

### Author Response · Authors · 2025-11-21

We sincerely thank all reviewers for their thoughtful, constructive, and detailed feedback. We are grateful for the time and expertise each reviewer invested in evaluating our submission. The comments significantly strengthened our work and helped us clarify the scope, motivation, and contributions of the paper.

In particular, we appreciate the reviewers’ recognition of the importance of this first systematic study on visual grounding in medical MLLMs, the value and the strong creation process of the VGMED dataset, and the effectiveness and generality of VGRefine.

We address all concerns in detail below, and we will incorporate the suggested clarifications, analyses and additional experiments in the revised version.

---

### Author Response · Authors · 2025-11-21

We have added the new figures, experiments, and dataset details to the end of the revised manuscript (p.50 to p.57, Sec. Q and R in supplementary). For convenience, we have also included the same material in a separate PDF titled **“HOW DO MEDICAL MLLMS FAIL? A STUDY ON VISUAL GROUNDING IN
MEDICAL IMAGES (SUPPLEMENTARY MATERIAL FOR REBUTTAL)”**, uploaded under Supplementary Material to facilitate the reviewers’ access.

---

### Author Response · Authors · 2025-12-02
**[Summary of Reviews and Discussion for AC] Part 1/2**

Dear Area Chair:

Thank you very much for handling our paper. For your quick reference, we provide a structured summary of the reviewers’ feedback and our response during the discussion phase.


$ $

Overview of the Reviews
===============

We received **two ratings of 6 (Reviewers 2uoh and Bv76) and one rating of 4 (Reviewer Qozi)**.

Notably, Reviewer Qozi’s detailed assessments rate our paper as **Soundness: 3 (good), Presentation: 3 (good), Contribution: 3 (good)**, and explicitly mention:

> “If the authors can address the above questions, the reviewer promises to raise the score.”



 Although Qozi has not responded before the cutoff date, **Qozi’s comments substantially overlap with those of Reviewer Bv76** (see Main Questions in part 2), who has already indicated that the questions have been addressed.

> “Overall, the authors have addressed my questions regarding the method’s motivation, interpretability, design choices, and potential training factors with sufficient and well-reasoned clarifications. The newly added ablation studies and more comprehensive visualization examples also strengthen the analysis. Moreover, the expanded cross-model comparisons and cross-domain analyses further enhance the credibility of the arguments.”



$ $


Positive Feedback from Reviewers
===============

We thank all the reviewers for their valuable time and effort in reviewing our work. We appreciate the Reviewers' positive rating and kind comments, such as:

- "The study is interesting. By examining whether MLLMs can reuse their reasoning attention for localization, the authors observe that models’ attention during question answering is not fully concentrated on relevant image regions, and finally propose VGRefine to improve model performance." (Reviewer Qozi)

- "The paper is **the first to systematically verify** that insufficient visual grounding is one of the key factors leading to poor performance of medical MLLMs." (Reviewer Qozi)

- "The proposed method achieves state-of-the-art results on six different Med-VQA benchmarks covering eight imaging modalities and more than 110,000 VQA samples." (Reviewer Qozi)

- "The study is **well-motivated, carefully designed, and supported by strong empirical results** across multiple tasks and modalities." (Reviewer 2uoh)

- "The paper isolates visual grounding as a distinct bottleneck in medical multimodal models, a perspective that is under-explored yet highly relevant for trustworthy deployment." (Reviewer 2uoh)

- "Layer- and head-specific analysis of cross-attention provides an interpretable pathway to understanding failure behavior, in line with mechanistic interpretability trends in multimodal research." (Reviewer 2uoh)

- “The inference-time attention refinement is conceptually clean, model-agnostic, and introduces no additional data or parameters. The performance gains across diverse modalities and benchmarks are substantial.” (Reviewer 2uoh)

- "The paper **identifies a clear and meaningful problem**—the weak visual grounding ability of medical MLLMs—and addresses it through a structured, quantitative framework." (Reviewer Bv76)

- "The proposed inference-time method, VGRefine, is concise, computationally efficient, and easy to deploy." (Reviewer Bv76)

- “The experimental section is extensive, including evaluations on multiple datasets, ablation studies, and expert assessments.” (Reviewer Bv76)

- “The combination of quantitative metrics and human evaluation adds credibility and depth to the analysis.” (Reviewer Bv76)


$ $


(continue in Part 2)

---

> ### Author Response · Authors · 2025-12-02
> **[Summary of Reviews and Discussion for AC] Part 2/2**
>
> Main Questions from Reviewers and Our Responses
> ===============
>
> We appreciate the reviewers for their constructive comments. Importantly, **no reviewer considers the paper to have critical issues**. Below, we summarize the main points from the reviews and our corresponding responses:
>
>
> >**Q**: More analysis to explain the causes behind the observed phenomenon. (Reviewer Qozi & Bv76)
>
> **A**: We highlighted a key potential reason for inadequate grounding on medical images: excessive intra-medical variability in how different imaging modalities are internally represented. Our quantitative analysis showed that modalities such as CT, MRI, X-ray, and ultrasound are embedded as highly separated, heterogeneous groups rather than as a coherent domain. Such excessive intra-medical variability makes it difficult for the model to learn a stable, unified mapping between textual concepts and visual features. (details in Reviewer Qozi Part 1 & Bv76 Part 5)
>
> $ $
>
>
> >**Q**: Whether the observed grounding difficulty is model-specific or domain-specific? Can the grounding difficulty be observed on more advanced general domain models e.g., Qwen2.5-VL or InternVL3? (Reviewer Qozi & Bv76) Can the proposed VGRefine hold its effectiveness on more advanced general domain models? (Reviewer Qozi)
>
> **A**:
> We clarified that we have already included in our experiments the latest **medical** MLLMs, which is the focus of this paper. Most of the medical MLLMs in our analysis/experiments were proposed in less than a year.
>
> We clarified that, in the main paper, we have already presented experiments to show that the same medical MLLMs that exhibit poor grounding on medical images demonstrate clear and coherent grounding when applied to natural RGB images, which strongly supports our main finding: **the grounding deficiency is not due to model weakness, but rather that medical images introduce domain-inherent grounding challenges that current medical MLLMs do not yet adequately address.** (details in Reviewer Qozi Part 2,3 & Bv76 Part 4)
>
> Furthermore, we follow reviewer’s suggestion to include the latest general domain MLLMs: Qwen2.5-VL and InternVL3. **The results further support our main finding:** visual grounding deficiency on medical images persists even in the latest general-purpose models. (details in Qozi Part 3 and Bv76 Part 4)
>
> In addition, the effectiveness of the proposed method (VGRefine) still holds in these latest general models and the same conclusions can be obtained. (details in Reviewer Qozi Part 3)
>
> $ $
>
>
> >**Q**: Explain why using only the last input text token to compute attention maps? (Reviewer Bv76)
>
> **A**: We clarified that using the last input text token to compute attention maps follows standard practice in prior visual-grounding work on general-domain MLLMs, enabling direct comparison with our finding that grounding often fails in medical images. To validate this design, we conducted an ablation study using alternative attention-map constructions and found that the last-token approach achieves equal or better alignment with ground-truth regions. (details in Reviewer Bv76 Part 2)
>
> $ $
>
>
> >**Q**: “The benchmark curation process is strong, but clearer quantitative annotation statistics (e.g. agreement checks, diversity distribution) would further support dataset rigor.” (Reviewer 2uoh)
>
> **A**: We provided detailed quantitative annotation statistics following reviewer suggestion (details in 2uoh Part 2)
>
>
> $ $
>
>
> Additionally, during the discussion phase, we clarified points that were overlooked or misinterpreted by reviewers and added further detail where needed. We specified that we did not claim grounding failures never occur in other specialized domains (Reviewer Qozi, Part 4), and explained that our head/layer choices are data-driven (Reviewer 2uoh, Part 3). Finally, we clarified that weak visual grounding is not presented as the sole cause of medical MLLM failure, but as one important, underexplored failure mode alongside semantic grounding and reasoning issues (Reviewer Bv76, Part 1).
>
>
> $ $
>
>
> We submitted our responses and updated manuscript on 21-Nov, carefully addressing all reviewers’ questions. We have added new figures, experiments, and dataset details to the end of the revised manuscript. We have also included the same material in a separate PDF titled **“Supplementary Material for Rebuttal”**, uploaded under Supplementary Material to facilitate your access.
>
>
> $ $
>
>
> We sincerely appreciate your time and effort in overseeing the review process.
>
> Regards, Authors

---

### Meta-Review · Area_Chair_yKma · 2025-12-29

**Summary:**

This paper investigates visual grounding failures in medical multimodal large language models (MLLMs), introduces a clinician-curated diagnostic benchmark (VGMED), and proposes a lightweight inference-time attention refinement method (VGRefine). The reviewers generally found the problem important, the diagnostic perspective novel, and the empirical study thorough. However, several concerns influenced the evaluation:

(1) Explanatory depth: Multiple reviewers (notably Reviewer Qozi and Reviewer Bv76) raised concerns that the paper is primarily diagnostic and descriptive, with limited mechanistic or mathematical explanation for why visual grounding fails in medical domains.

(2) Causal claims vs. evidence: Reviewers questioned whether the observed grounding failures can be causally attributed to visual grounding alone, as opposed to semantic grounding or reasoning deficiencies. The need to clearly disentangle these factors was emphasized.

(3) Model coverage and recency: Some reviewers were concerned that the main experiments relied on relatively older models and requested validation on newer, stronger general-purpose and medical MLLMs.

Overall, while reviewers acknowledged the contribution and empirical rigor, the paper was initially perceived as being slightly below the acceptance threshold due to concerns about explanatory strength.

**Reviewer Concerns:**

Most reviewer concerns were satisfactorily addressed in the rebuttal, including questions about domain specificity, model recency, methodological design choices, and benchmark rigor. Additional cross-domain and cross-model experiments clarified that the observed grounding failures are specific to medical images rather than particular model architectures, and that the phenomenon persists in newer state-of-the-art MLLMs. The authors also provided sufficient justifications and ablations for key design decisions and strengthened confidence in the VGMED benchmark through more detailed annotation statistics.

Concerns still outstanding
(1) The paper still lacks a sufficiently strong theoretical or mechanistic explanation for why visual grounding fails in medical MLLMs. While the authors propose intra-medical modality heterogeneity as a plausible factor and support it with MIR-based analysis, the evidence remains largely correlational and operates at the modality level rather than at the concept or region level. As a result, the work provides limited causal understanding of how representation fragmentation leads to grounding failure.

(2) The proposed VGRefine method primarily serves as an inference-time mitigation rather than a mechanism that fundamentally improves model representations or learning dynamics. Although effective and practical, it does not directly address the underlying causes of grounding failure during training, which limits the depth of the contribution from a modeling perspective.

(3) Despite strengthened empirical coverage, the paper remains more diagnostic than explanatory, emphasizing empirical characterization and mitigation over deeper theoretical insights into grounding behavior in medical multimodal models.

**Reviewer Scores:**

Reviewer Qozi (initial rating: 4 / marginally below acceptance threshold) Although the rebuttal addressed many of the reviewer’s specific questions through additional experiments and clarifications, the core concerns regarding explanatory depth and causal understanding were not fully resolved. While the reviewer indicated openness to increasing the score if these issues were addressed, the rebuttal primarily strengthened empirical coverage rather than providing deeper theoretical insight. As a result, the score would likely remain unchanged or increase only marginally, but still below the acceptance threshold.

Reviewer 2uoh (initial rating: 6 / marginally above acceptance threshold) the reviewer would likely maintain the original score.

Reviewer Bv76 (initial rating: 6 / marginally above acceptance threshold) Reviewer Bv76 explicitly stated in the post-rebuttal discussion that, despite appreciating the additional analyses and clarifications, they decided to maintain their original rating. This confirms that the rebuttal did not materially change the reviewer’s overall assessment of the paper’s contribution and limitations.

---

### Decision · Program_Chairs · 2026-01-26

Accept (Poster)